# Towards Continual Knowledge Learning of Language Models

**Joel Jang**[1]  **Seonghyeon Ye**[1]  **Sohee Yang**[1]  **Joongbo Shin**[2]
**Janghoon Han**[2]  **Gyeonghun Kim**[2]  **Stanley Jungkyu Choi**[2]  **Minjoon Seo**[1]
[1]KAIST AI  [2]LG AI Research
{joeljang,vano1205,sohee.yang,minjoon}@kaist.ac.kr
{jb.shin,janghoon.han,ghkayne.kim,stanleyjk.choi}@lgresearch.ai

## ABSTRACT

Large Language Models (LMs) are known to encode world knowledge in their parameters as they pretrain on a vast amount of web corpus, which is often utilized for performing knowledge-dependent downstream tasks such as question answering, fact-checking, and open dialogue. In real-world scenarios, the world knowledge stored in the LMs can quickly become outdated as the world changes, but it is non-trivial to avoid catastrophic forgetting and reliably acquire new knowledge while preserving invariant knowledge. To push the community towards better maintenance of ever-changing LMs, we formulate a new continual learning (CL) problem called Continual Knowledge Learning (CKL). We construct a new benchmark and metric to quantify the retention of time-invariant world knowledge, the update of outdated knowledge, and the acquisition of new knowledge. We adopt applicable recent methods from literature to create several strong baselines. Through extensive experiments, we find that CKL exhibits unique challenges that are not addressed in previous CL setups, where parameter expansion is necessary to reliably retain and learn knowledge simultaneously. By highlighting the critical causes of knowledge forgetting, we show that CKL is a challenging and important problem that helps us better understand and train ever-changing LMs. The benchmark datasets, model checkpoints, and code to reproduce our results are available at this https URL.

## 1  INTRODUCTION

Recent works have shown that large Language Models (LM), such as T5 (Raffel et al., 2019) and GPT-3 (Brown et al., 2020), have the capability of storing a tremendous amount of world knowledge in their parameters when pretrained on a vast corpus of text (Petroni et al., 2019). These pretrained LMs have shown potential to serve as knowledge bases when probed for world knowledge without any finetuning through the LAnguage Model Analysis (LAMA) task (Petroni et al., 2019), which requires probing LMs for world knowledge in a zero-shot manner through slot-filling, and promising results utilizing the encoded world knowledge when finetuned on various Knowledge Intensive Language Tasks (KILT) (Petroni et al., 2021), e.g., question answering, knowledgeable open dialogues.

While the world knowledge stored in LMs has diverse use cases, it can quickly become outdated as the world changes fast, and LMs need to frequently renew their internal world knowledge accordingly. For example, it is impossible to probe for ***new*** information such as "______ *won the US Election 2020*" from the original T5 (Raffel et al., 2019) which was pretrained on C4 web corpus from April 2019.[1] Also, information that may have once been considered accurate may no longer be valid because the information has been ***updated***. For instance, the answer to "*Which soccer team does Cristiano Ronaldo play for?*" has changed from *Juventus* to *Manchester United* in September 2021. Meanwhile, ***time-invariant*** information learned from the original corpus such as "*Barack Obama was born in Honolulu, Hawaii*" should not be altered within the LMs.

---

[1]T5 was initially pretrained on the C4 dataset (about 750 GB), which is a cleansed dump of Common Crawl extracted from the web in April 2019.

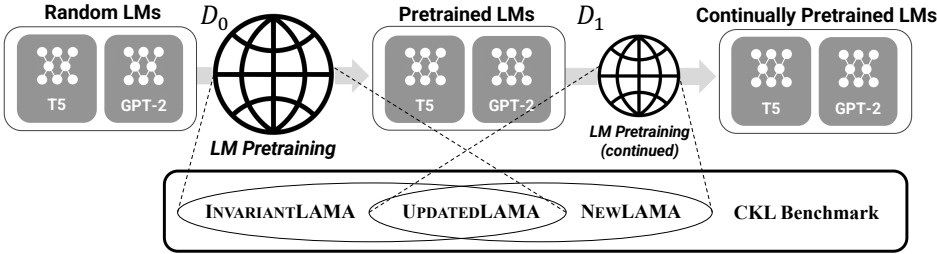

Figure 1: Overview of the CONTINUAL KNOWLEDGE LEARNING benchmark. INVARIANTLAMA is used to measure the *time-invariant* world knowledge gained from $D_0$. UPDATEDLAMA is used to measure the *update* of world knowledge from $D_0 \rightarrow D_1$. NEWLAMA is used to measure *new* world knowledge gained from $D_1$.

Despite its importance, the challenge of renewing the internal world knowledge stored in the parameters of LMs is nontrivial and has only been explored in rather specific settings. For example, recent works have proposed to modify specific target knowledge such as individual facts (De Cao et al., 2021; Zhu et al., 2020; Dai et al., 2021). Dhingra et al. (2021) have addressed LMs as temporal knowledge bases by jointly modeling text with its timestamp. But the problem of renewing the world knowledge of LMs in a more general and scalable way, such as through continual pretraining on a corpus with new knowledge, has not been formally formulated or explored by previous works. Moreover, the community lacks a benchmark that can be used to systematically study how the internal knowledge of LMs changes through the training on new information. Lastly, methodologies to effectively renew the knowledge of LMs at scale have yet to be thoroughly explored.

In this work, we propose a novel continual learning (CL) formulation named CONTINUAL KNOWLEDGE LEARNING (CKL), where we attempt to renew the internal world knowledge of LMs through continual pretraining on new corpora. We systematically categorize world knowledge into three main categories and make benchmark datasets to measure each of them during CKL: (1) INVARIANTLAMA for *time-invariant* world knowledge in LMs that should not be forgotten or altered, (2) UPDATEDLAMA for outdated world knowledge that needs to be ***updated*** in the LMs, and (3) NEWLAMA for ***new*** world knowledge that should be injected into the LMs. We also propose a novel metric named FUAR (**F**ORGOTTEN / (**U**PDATED + **A**CQUIRED) **R**ATIO) that can measure the trade-off between forgetting, updating, and acquiring knowledge. Finally, while one might think of implementing contemporary CL methods for this benchmark, we show that CKL has nontrivial differences to traditional CL formulations and require approaches specific to CKL. We find and compare model architectures and training methodologies (Chen et al., 2020; He et al., 2021; Hu et al., 2021; Wang et al., 2021b) from the literature that have shown potential to mitigate forgetting of knowledge gained during pretraining, establishing them as baselines for the CKL benchmark.

In sum, while the challenge of renewing the internal world knowledge of LMs is essential in real-world scenarios, it has yet to be formulated or extensively explored. Therefore, in this paper:

- We propose a novel CL formulation called CONTINUAL KNOWLEDGE LEARNING (CKL) and construct a new benchmark to measure the amount of forgetting and amount of world knowledge gained by continued pretraining on a novel language modeling corpus that we construct, containing new knowledge.

- We explore LM architectures and training methodologies that are natural baselines for CKL in literature, denoting them as CKL methods, and performing extensive experiments on our CKL benchmark. We categorize them into regularization, rehearsal, and parameter-expansion methods, same as in traditional CL literature, and compare the effectiveness of each type of method using a novel metric named FUAR that we propose to measure the trade-off between forgotten knowledge and updated or acquired knowledge.

- Towards creating an ever-changing LM, we perform extensive analysis in the CKL benchmark and highlight important challenges and findings: parameter-expansion methods have the limitation of memory inefficiency despite performing the best in most of our experiments and seeing the same data repeatedly during continued pretraining is a critical cause of forgetting. Also, we show interesting results that need further exploration: learning rate can be varied to balance the forgetting and learning of new knowledge, CKL may help in

performing previous-knowledge-intensive tasks after gaining new world knowledge, and CKL methods are transferable across LM architectures despite showing a different trend in performance.

An overview of the proposed CKL benchmark is shown in Figure 1.

## 2    RELATED WORK

Language Models (LMs) utilizing knowledge from external sources, such as Retrieval-Augmented Generation (RAG) (Lewis et al., 2020a) and Blender Bot 2.0 (Xu et al., 2021; Komeili et al., 2021), cope with the changing world by updating the external sources during inference or searching the internet for retrieving recent information. However, recent works have shown that these memory-augmented models suffer from *hallucination*, which means that they present false information as if it were correct, despite being given updated knowledge during inference (Zhang & Choi, 2021), which worsens as the size of the LM increases (Longpre et al., 2021), making it more so important for implicit parameters to be renewed as well.

In order to renew the internal knowledge of LMs, one might consider pretraining LMs from scratch with a newly updated text corpus of a scale similar to the one used during initial pretraining, such as a recent dump of the entire Wikipedia. However, this approach is computationally demanding and also environmentally harmful (Patterson et al., 2021). Another alternative approach is continuing the pretraining process on a much smaller corpus containing new world knowledge, but such a methodology is known to suffer from *catastrophic forgetting* (McCloskey & Cohen, 1989; Kirkpatrick et al., 2017), where the models forget previously learned knowledge as they acquire new knowledge.

Lazaridou et al. (2021); Jin et al. (2021) suggests implementing prior Continual Learning (CL) methods (Sun et al., 2020; d'Autume et al., 2019) to address this problem. However, it is important to note that there are nontrivial differences between traditional CL and the proposed Continual Knowledge Learning (CKL) formulation which make applying traditional CL methods inadequate. In traditional CL, methods can be largely categorized into *regularization*, *rehearsal*, and *parameter-expansion* methods. (1) While regularization methods (Kirkpatrick et al., 2017) require identifying important parameters used for previous tasks, exactly how and where the knowledge is stored in the parameters of an LM is currently extremely difficult to identify and localize (Vig et al., 2020; De Cao et al., 2021). (2) While prior rehearsal methods (Lopez-Paz & Ranzato, 2017) consider learning all of the streams of tasks at once (multi-task learning) as the performance upper-bound and replicate such a setting with samples stored in the episodic memory, a few samples from the pretraining corpus cannot represent the overall world knowledge from the corpus. Moreover, if LMs are pretrained on a shuffled concatenation of stream of corpora, there is no guarantee that the LMs will acquire the correct, recent information from the recent corpora, especially in cases where the former corpora are much bigger than the latter ones, which is shown by experiments in Section 5.1. (3) Lastly, prior parameter-expansion methods (Rusu et al., 2016; Yoon et al., 2018) focus on *learning a stream of different tasks via strong supervision*, while in CKL, the focus is *constantly updating world knowledge from a stream of corpora via self-supervision*.

Because of these fundamental differences, instead of contemporary CL methods mentioned above, we explore methodologies from the literature that are suitable for CKL (Chen et al., 2020; He et al., 2021; Hu et al., 2021; Wang et al., 2021b), modifying and adapting each method according to our needs as CKL methods. Lastly, while it has been pointed out that some of the traditional CL formulations may have little practical importance in real-world scenarios by Prabhu et al. (2020), CKL is much closer to the initial motivation behind CL, which is that the "fundamental characteristic of natural intelligence is its ability to continually learn new knowledge while updating information about the old ones" (Prabhu et al., 2020). Details of related works regarding the traditional CL methods and how CKL methods address the fundamental differences are provided in Appendix A.

## 3    CONTINUAL KNOWLEDGE LEARNING (CKL)

In this section, we explain the formulation of the task, the data construction process, and the proposed metric measuring the trade-off between forgetting previous world knowledge and updating and learning of new world knowledge.

## 3.1 TASK FORMULATION

When viewing the task of renewing the internal knowledge of LMs as one of CL formulations, pretraining on the original corpus can be considered as a *previous task*, and continued pretraining on new corpus can be considered as the *current task*, the main objective becoming retaining the *time-invariant* world knowledge gained through initial pretraining while efficiently learning *new* and *updated* world knowledge through continued pretraining. Throughout the paper, we let $D_0$ refer to the corpus used for initial pretraining and let $D_1$ denote the new corpus used for continued pretraining.

**New Text Corpus for Language Modeling**  For LMs to renew their internal knowledge, they need to be continually pretrained on a new text corpus $D_1$ which has the updated and new information. $D_1$ should ideally be much smaller than $D_0$, as a large $D_1$ amounting to the size of $D_0$ will result in massive computational costs similar to pretraining the LMs from scratch. For constructing $D_1$, we crawl recently published news articles from the web making CC-RECENTNEWS.[2]

**Probing LMs for World Knowledge**  The most widely used task for probing LMs for world knowledge is the LAnguage Model Analysis (LAMA) (Petroni et al., 2019) task, which consists of cloze sentences created from a set of knowledge sources using manually defined templates. We define that an LM *knows* a fact if it can successfully predict in a zero-shot manner the masked entity in the cloze sentence, such as "*Dante was born in _____*" as *Florence*. While there may be other alternatives for measuring the world knowledge encoded in LMs[3], we construct our main datasets as LAMA tasks, while also additionally providing the corresponding question pairs to the cloze sentences for those who want to test on CBQA as well.

**Measuring Retention of Time-invariant World Knowledge**  We define *time-invariant* world knowledge as the information present in $D_0$ that has no possibility of conflicting with information from $D_1$. For example, if the information of the *birthplace of Barack Obama* is present in $D_0$, it is unlikely that $D_1$ contains information that contradicts that fact. Also, we classify instances where the time-stamps are fixed such as "*Cristiano Ronaldo played for _____ in 2010.*" as *time-invariant*. These *time-invariant* instances should not be changed as LMs are continually pretrained on $D_1$. In order to measure how much *time-invariant* information is lost due to *catastrophic forgetting* during continued pretraining, we create INVARIANTLAMA, a subset of LAMA (Petroni et al., 2019), consisting of only *time-invariant* cloze sentences detailed in Appendix B.1.

**Measuring Update of Outdated World Knowledge**  In this work, we define *outdated* world knowledge as information that is conflicting between $D_0$ and $D_1$. For example, the President of the US may be *Barack Obama* in $D_0$ and *Joe Biden* in $D_1$. In this case, the LM should update its internal knowledge as *Joe Biden* as the US president. If an LM is pretrained on both $D_0$ and $D_1$ simultaneously, there is no guarantee that the LM will acquire the correct, recent information from $D_1$, especially in cases where $D_0$ is much bigger than $D_1$, which is one of the biggest difference between the CKL and traditional CL setting. For measuring *update* of outdated information, we construct UPDATEDLAMA which is made up of cloze statements for which answers can be found in both $D_0$ and $D_1$, but are conflicting.

**Measuring Acquisition of New World Knowledge**  We define *new* world knowledge as the information present in $D_1$, but not in $D_0$. To measure *new* knowledge acquired through continued pretraining on $D_1$, we construct NEWLAMA which is made up of detailed cloze statements requiring *new* knowledge from $D_1$ to correctly answer. We provide two datasets for measuring *new world knowledge*: NEWLAMA, for which each of the instances is verified that the answer does not exist in $D_0$, but only in $D_1$, and NEWLAMA-EASY for which each of the instances does not perfectly comply with our strict definition of *new* world knowledge due to its creation process, but is used to generally measure the new knowledge acquired from continued pretraining on $D_1$ at a larger scale.

---

[2]CC-RECENTNEWS consists of 221,779 articles (~168M tokens), which is estimated to be about 750 times smaller than C4, a cleansed version of the April 2019 Common Crawl dataset (https://commoncrawl.org/) that was used to initially pretrain the T5 LM (Raffel et al., 2019).

[3]Closed-book question answering (CBQA) (Roberts et al., 2020) can also be considered as a task that measures the world knowledge of LMs through finetuning, but it has been pointed out that much of its performance increases are due to the test-train overlap (Lewis et al., 2020b; Wang et al., 2021a) in the datasets.

Table 1: Dataset statistics. Input and answer length are the corresponding average token lengths.

| Dataset | Size | Input Length | Answer Length | Dataset | Size | Input Length | Answer Length |
|---|---|---|---|---|---|---|---|
| INVARIANTLAMA | 17474 | 11.9 | 1.3 | NEWLAMA | 797 | 14.7 | 8.7 |
| UPDATEDLAMA | 924 | 13.7 | 9.4 | NEWLAMA-EASY | 11177 | 44.4 | 6.1 |

NEWLAMA-EASY can be considered *easier* since each instance was constructed to be similar to the data distribution seen during continued pretraining.

**Dataset Construction** The data for continual pretraining, CC-RECENTNEWS, is constructed using news-please (Hamborg et al., 2017). INVARIANTLAMA is constructed by manually selecting 28 *time-invariant* relations from T-Rex (Elsahar et al., 2018). For UPDATEDLAMA and NEWLAMA, we use Amazon Mechanical Turk (mturk)[4] for crowd-sourcing Human Intelligent Tasks (HITs). The process requires selecting answerable questions from a list of questions generated by the model introduced in Lewis et al. (2021) and converting them into cloze sentences. We have also separately hired 11 experts to verify the correctness and search the C4 database to categorize each instance following our definition of *updated* and *new*. NEWLAMA-EASY is constructed at a larger scale through a two-phase mturk process where sentences selected from articles containing new information are decontextualized and paraphrased[5] before being masked, verified and converted to corresponding questions. The constructed dataset statistics are in Table 1. Important details about the data construction pipeline, examples, and more fine-grained statistics are provided in Appendix B.

## 3.2 COMBINED METRIC FOR CKL

We propose a novel metric, **FUAR** (**F**ORGOTTEN / (**U**PDATED + **A**CQUIRED) **R**ATIO), that can compare the efficiency of each CKL method using the trade-off between forgotten time-invariant knowledge and updated or newly acquired knowledge. FUAR represents relatively *how many* time-invariant knowledge instances are forgotten in order to learn *one* new or updated knowledge instance. We first define FUAR for the general case where there can be multiple corpora used for training an ever-changing LM.

Let $T$ be an arbitrary task and $(D_i)_{i=0}^n$ be a sequence of corpora used for LM pretraining, where $D_0$ is the initial pretraining corpus. We define $\text{Gap}(T, D_a, D_b) = Score(T)$ of $LM_a - Score(T)$ of $LM_b$, where $LM_a$ represents the LM after being pretrained on $D_a$. Then, we denote $\mathbb{T}^F = (T_i^F)_{i=0}^{n-1}$ as a sequence of tasks from $(D_i)_{i=0}^{n-1}$ measuring the forgetting of invariant-knowledge from each corresponding corpus. If there is no such task from corpus $D_i$, the value of $T_i^F$ is set to *n.d.*, which means *not defined*. Likewise, we denote $T_n^U$ and $T_n^A$ as tasks from $D_n$ measuring the *update* and *acquisition* of new knowledge, respectively. We define FUAR as follows:

$$\text{FUAR}(\mathbb{T}^F, T_n^U, T_n^A) = \begin{cases} \dfrac{\sum_{i=0}^{n-1} \max(0, \text{Gap}(T_i^F, D_i, D_n)) \mathbb{1}_{\{T_i^F \neq n.d.\}}}{\sum_{i=0}^{n-1} \{\max(0, \text{Gap}(T_n^U, D_n, D_i)) \mathbb{1}_{\{T_i^F \neq n.d.\}} + \max(0, \text{Gap}(T_n^A, D_n, D_i)) \mathbb{1}_{\{T_i^F \neq n.d.\}}\}}, \\ \qquad\qquad \text{if denominator} > 0, \\ no\ gain, \text{ otherwise.} \end{cases}$$

(1)

The choice of benchmark tasks $\mathbb{T}^F$, $T_n^U$, and $T_n^A$ can differ according to each experimental setup. FUAR value of 1.0 represents an equal trade-off scenario where *one* time-invariant knowledge instance of $\mathbb{T}^F$ is forgotten on average to gain one new or updated knowledge instance of $T_n^U$ and $T_n^A$. The two terms in the denominators are summed because newly gained knowledge and updated knowledge are mutually exclusive by definition. When the value is smaller than 1, it means that the model obtains more new or updated knowledge than the amount of forgotten knowledge, so methods

---

[4]https://www.mturk.com

[5]Decontextualization model from Choi et al. (2021) and back-translation model from Tiedemann & Thottingal (2020) is used.

that exhibit a low FUAR value can be considered suitable for CKL. If the value is zero, then it is a case where no forgetting occurs at all and is the upper bound for performance. If the denominator is 0, we denote the case as *no gain* and regard it as the worst possible case.[6]

## 4    EXPERIMENTAL SETUP

We perform extensive experiments with an encoder-decoder model, T5 (Raffel et al., 2019), a large LM (~ 737M params) initially pretrained on April 2019 dump of C4 and May 2020 dump of Wikipedia (thus $D_0$ in our experiments) with salient span masking (SSM). The details of the pretraining, continual pretraining, and evaluation configurations are in Appendix C. We establish the following methods as the baselines for the CKL benchmark and categorize them into *regularization*, *rehearsal*, and *parameter-expansion* methods. The specific hyperparamters used for the implementation of each method are detailed in Appendix D.

**Initial** refers to the setting where we evaluate the LM before any continued pretraining. The performance of this model can be considered as the *upper-bound* for INVARIANTLAMA and *lower-bound* on UPDATEDLAMA and NEWLAMA.

**Vanilla** is a specific setting of further pretraining (Gururangan et al., 2020), where the domain is *new* knowledge, and the LM is further pretrained without any training strategies.

**RecAdam** (Chen et al., 2020) falls into the category of regularization methods. It places a stronger independent assumption among the model parameters than the traditional regularization method (EWC (Kirkpatrick et al., 2017)) and does not access the initial pretraining corpus to regularize the model weights during continued pretraining. The optimizer is annealed so that less regularization is applied as the training progresses.

**Mix-Review** (He et al., 2021) falls into the category of rehearsal methods, which assumes access to the initial pretraining corpus and mixes in random subsets of the initial pretraining data during continued pretraining, depending on the mix-ratio at the current time step. As the training progresses, the mix-ratio decays towards 0, decreasing the amount of the mixed original data at each iteration.

**LoRA** (Hu et al., 2021) falls into the category of parameter-expansion methods. It freezes the original parameters of the LM and adds trainable rank-decomposition matrices into each layer that are updated during continued pretraining. Hu et al. (2021) has implemented this approach with decoder-only models (GPT-2 (Radford et al., 2019) & GPT-3 (Brown et al., 2020)) while we apply it to an encoder-decoder model, denoting it as T5-LoRA.

**K-Adapter** (Wang et al., 2021b) is another parameter-expansion method that freezes the original parameters of the LM while adding $k$ number of new layers, namely *adapters*, that are updated during continued pretraining. Wang et al. (2021b) have shown successful injection of *factual* and *linguistic* knowledge for encoder-only models, BERT (Devlin et al., 2019) & RoBERTa (Liu et al., 2019), while we also apply it to an encoder-decoder model, T5, and decoder-only model, GPT-2.

**Modular** is a newly proposed parameter-expansion method specifically for encoder-decoder models which freezes the original, pretrained encoder while adding a new, randomly initialized encoder that is updated during continued pretraining. For the newly added encoder, we vary the size to *T5-small* while keeping the size of the original encoder and decoder to be *T5-large*.

## 5    EXPERIMENTAL RESULTS

In this section, we first show the main experimental results for the CKL Benchmark. Then, since multiple steps of continual knowledge learning, i.e., CKL are needed for training a true, ever-changing LM, we explore the effects of multiple CKL phases as well as how epochs, corpus size, and the total number of training steps affect CKL. We further explore how learning rates affect CKL in Appendix E, how continual pretraining on $D_1$ affects the performance of KILT tasks which re-

---

[6]Each of the last two sentences means that we do not measure positive *backward* transfer and negative *forward* transfer, respectively. The latter in some cases actually do happen (shown in Appendix G). Explanations about the backward and forward transfer are in Appendix A.1.

Table 2: Zero-shot probing performance on the CKL benchmark. The best results for each task and metric are shown in bold, and the second-best results are underlined.

| Method | # of Params (Trainable / Total) | IL EM | UL EM | NL EM | NLE EM | FUAR $((\mathbf{IL}), \mathbf{UL}, \mathbf{NL}) \downarrow$ |
|---|---|---|---|---|---|---|
| T5-Initial | 0M / 737M | **24.17** | 1.62 | 1.88 | 10.32 | - |
| T5-Vanilla | 737M / 737M | 12.89 | 10.17 | 3.77 | 17.75 | 1.08 |
| T5-RecAdam | 737M / 737M | 13.20 | 12.55 | 4.02 | 17.85 | 0.84 |
| T5-MixReview | 737M / 737M | 13.92 | 6.49 | 2.89 | 14.86 | 1.74 |
| T5-LoRA | 403M / 738M | 16.58 | **12.77** | 4.52 | **19.56** | 0.55 |
| T5-Kadapters (k=2) | 427M / 762M | 19.59 | 12.34 | **5.03** | 18.75 | 0.33 |
| T5-Kadapters (k=3) | 440M / 775M | 19.76 | 12.66 | 4.02 | 19.00 | 0.33 |
| T5-Modular | 438M / 773M | 20.29 | 12.66 | 4.65 | 19.24 | **0.28** |

quire knowledge from $D_0$ in Appendix F, how CKL methods transfer across LM architectures in Appendix G, and how the prediction outputs change during CKL in Appendix H.

## 5.1 MAIN RESULTS

Table 2 shows our main experimental result on the CKL benchmark. While only the exact match (EM) is reported in Table 2, we report the F1 score as well as the mean precision at k ($P@k$, k=1,5,10,20,50,100) in Appendix J. The T5 models are originally pretrained on C4 (about 1 trillion token updates) and Wikipedia, which is considered as $D_0$.[7], and then continually pretrained on CC-RecentNews (corpus $D_1$) for 4 epochs (25k global training steps, about 673 million token updates) using each of the CKL methods. Each of IL, UL, NL, NLE stands for INVARIANTLAMA, UPDATEDLAMA, NEWLAMA, and NEWLAMA-EASY, respectively. Detailed descriptions about the setup for this experiment are included in the caption.

We first find that all of the CKL methods except for T5-MixReview are more effective at forgetting less time-invariant knowledge while updating and acquiring new knowledge than using the naïve approach of T5-Vanilla as shown by the FUAR. This result also highlights the main difference between CKL and CL; while rehearsal methods show strong performances in traditional CL settings (Prabhu et al., 2020; Bang et al., 2021), in CKL, it shows the worst performance since the update of outdated knowledge and acquisition of new knowledge is severely deterred as shown in the performance of UL and NL while not showing competitive mitigation of forgetting as shown in the performance of IL compared to other CKL methods. Amongst the other CKL methods, we observe a rather consistent trend that the parameter-expansion methods achieve better results. The first and second-best results on all of UL, NL, and NLE are all from parameter-expansion methods. Meanwhile, although UL and NL are constructed following the same procedure, there is a huge difference between the EM scores of UL and NL. We analyze the source of this difference in Appendix I.

Figure 9 visualizes how the EM scores of each task change as T5-Kadapters, the CKL method with the most robust performance, and T5-Vanilla are continually pretrained on $D_1$. In all of the tasks, the performance of T5-Initial can be considered as the upper-bound for IL and lower-bound for UL, NL, NLE. Corresponding with our main observations, CKL allows considerable retention of *time-invariant* world knowledge while improving updating and gaining new world knowledge compared to T5-Vanilla, mitigating the overall trade-off.

## 5.2 EXPLORING MULTIPLE PHASES OF CKL

In order to show the potential for creating a truly ever-changing LM, we explore the effect of multiple CKL phases by creating CC-RECENTNEWS-SMALL, denoted as SMALL, which is a small variant of CC-RECENTNEWS that consists of randomly sampled 10% of the original corpus. We then split

---

[7]In this work, we see C4 and Wikipedia together as $D_0$, because we do not measure how the knowledge in LMs change in between training on those two corpora.

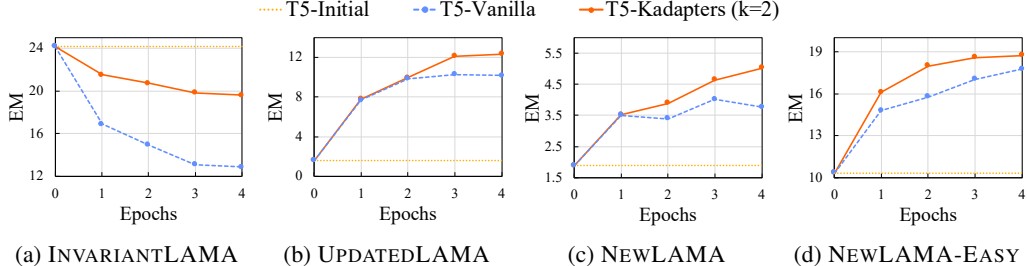

(a) INVARIANTLAMA    (b) UPDATEDLAMA    (c) NEWLAMA    (d) NEWLAMA-EASY

Figure 2: Performance at each epoch during continued pretraining in the main experimental setting.

Table 3: Zero-shot probing performance after T5 models are continually pretrained on different subsets of CC-RECENTNEWS. NLE and IL stand for NewLAMA-Easy and InvariantLAMA, respectively. There are three scenarios according to the corpus used for continual pretraining, explained in the text of Section 5.2. The FUAR of the three scenarios is calculated differently, and the corresponding tasks are shown in the table as the parameters of FUAR: $\mathbb{T}^F$, $T_n^U$, and $T_n^A$. In this setting, $\mathbb{T}^F$ consists of only a single task $T_0^F$ (IL) measuring the time-invariant information lost from $D_0$ only. For SMALL, we calculate the gap on NLE using the weighted sum of the gaps on $\text{NLE}_{\text{P1}}$ and $\text{NLE}_{\text{P2}}$ with uniform weights.

| Corpus | Method | # of Params (Trainable / Total) | IL EM | $\text{NLE}_{\text{P1}}$ EM | $\text{NLE}_{\text{P2}}$ EM | |
|---|---|---|---|---|---|---|
| | | | | | | **FUAR** |
| | T5-Initial | 0M / 737M | **24.17** | 8.69 | 9.45 | $((\mathbf{IL}), \boldsymbol{n.d.}, \mathbf{NLE}) \downarrow$ |
| SMALL (SMALL-P1 + SMALL-P2) | T5-Vanilla | 737M / 737M | 11.86 | 17.77 | 16.42 | 1.53 |
| | T5-RecAdam | 737M / 737M | 11.85 | 16.46 | 13.93 | 2.01 |
| | T5-MixReview | 737M / 737M | 14.36 | 14.18 | 13.93 | 1.97 |
| | T5-LoRA | 403M / 738M | 14.26 | 20.60 | 19.90 | 0.87 |
| | T5-Kadapters (k=2) | 427M / 762M | 18.16 | 18.34 | 16.42 | 0.72 |
| | T5-Kadapters (k=3) | 440M / 775M | 17.12 | **20.98** | **20.39** | **0.61** |
| | T5-Modular | 438M / 773M | 16.40 | 19.47 | 19.90 | 0.73 |
| | | | | | | **FUAR** |
| | T5-Initial | 0M / 737M | **24.17** | 8.69 | 9.45 | $((\mathbf{IL}), \boldsymbol{n.d.}, \mathbf{NLE_{P1}}) \downarrow$ |
| SMALL-P1 | T5-Vanilla | 737M / 737M | 9.68 | 20.60 | *11.44* | 1.22 |
| | T5-RecAdam | 737M / 737M | 11.78 | 20.42 | *11.94* | 1.06 |
| | T5-MixReview | 737M / 737 M | 16.13 | 15.88 | *11.94* | 1.12 |
| | T5-LoRA | 403M / 738M | 14.75 | **20.79** | *13.93* | 0.78 |
| | T5-Kadapters (k=2) | 427M / 762M | 19.11 | 20.60 | *10.95* | **0.42** |
| | T5-Kadapters (k=3) | 440M / 775M | 19.08 | 18.15 | *10.94* | 0.54 |
| | T5-Modular | 438M / 773M | 17.08 | 18.90 | *11.94* | 0.69 |
| | | | | | | **FUAR** |
| | T5-Initial | 0M / 737M | **24.17** | 8.69 | 9.45 | $((\mathbf{IL}, \boldsymbol{n.d.}), \boldsymbol{n.d.}, \mathbf{NLE_{P2}}) \downarrow$ |
| SMALL-P1→ SMALL-P2 | T5-Vanilla | 737 M / 737 M | 9.40 | 14.37 | **23.38** | 1.06 |
| | T5-RecAdam | 737M / 737M | 7.25 | 14.56 | 20.90 | 1.48 |
| | T5-MixReview | 737M / 737M | 13.20 | **17.20** | 16.92 | 1.47 |
| | T5-LoRA | 404M / 740M | 13.25 | 16.07 | 22.39 | 0.84 |
| | T5-Kadapters (k=2) | 427M / 788M | 15.78 | 16.07 | **23.38** | **0.60** |
| | T5-Kadapters (k=3) | 440M / 813M | 15.47 | 15.31 | 20.90 | 0.76 |
| | T5-Modular | 438M / 809M | 14.66 | 15.31 | 20.40 | 0.87 |

CC-RECENTNEWS-SMALL into two different splits by the published date of each article to simulate a setting where multiple CKL phases are needed, denoted as SMALL-P1 (05.2020 - 11.2020)) and SMALL-P2 (11.2020 - 04.2021). NLE[8] is also split into two different, smaller datasets, $\text{NLE}_{\text{P1}}$ and $\text{NLE}_{\text{P2}}$, each comprising of instances constructed from articles in SMALL-P1 and SMALL-P2,

---

[8]We use NEWLAMA-EASY instead of NEWLAMA because the number of instances in NL corresponding to articles from SMALL is too small for robust evaluation.

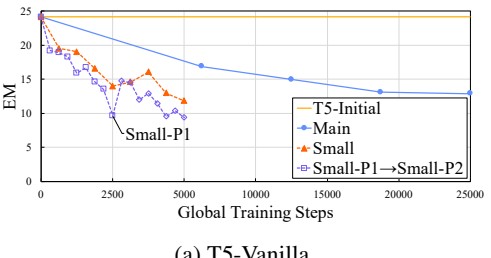

(a) T5-Vanilla

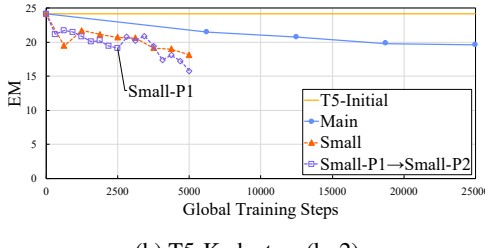

(b) T5-Kadapters (k=2)

Figure 3: Performance at each epoch on INVARIANTLAMA during continued pretraining in MAIN, SMALL, and SMALL-P1→SMALL-P2 scenarios. Each marker indicates the result at each continual pretraining epoch.

respectively. We compare how CKL methods for T5 perform on IL, NLE$_{P1}$, and NLE$_{P2}$ when continually pretrained entirely on SMALL for 5k steps (8 epochs), and when sequentially pretrained on SMALL-P1 and then on SMALL-P2 for 2.5k steps (8 epochs) each. In the scenario SMALL-P1→SMALL-P2, there are two CKL phases where $D_0$ is C4 and Wikipedia, $D_1$ is SMALL-P1, and $D_2$ is SMALL-P2. The rest of the configurations are set identical with the main experiments.

Comparing the performance on IL of the two scenarios, SMALL and SMALL-P1→SMALL-P2, results show that LMs are prone to more forgetting as they go through multiple CKL phases, despite having the same number of training steps. One of the reasons may be due to the learning rate scheduling, which is initialized at the start of each phase.

Furthermore, despite showing the best performance overall, the drawbacks of parameter-expansion methods are also highlighted in the SMALL-P1→SMALL-P2 setting; they require new parameters to be added at every phase of the update. For example, the number of total parameters of T5-Modular increases by 36M in every round of the continual pretraining phase. Likewise, considering a large number of CKL phases introduces new problems that should be additionally studied. Taking into account that LMs should be updated frequently with a small amount of data in real-world scenarios for gaining up-to-date world knowledge about the ever-changing world in a computation-effective manner, more research is needed to mitigate the amount of forgetting that follows the larger number of update phases.

**Effects of Epochs, Corpus Size, and Total Number of Training Steps in CKL on Forgetting** Figure 3 shows the result of T5-Vanilla and T5-Kadapters during continued pretraining in different scenarios from Table 2 and 3, where each point in the graph represents the performance of IL after every epoch. Comparing MAIN (4 epochs) and SMALL (8 epochs) in Figure 3 (a) T5-Vanilla, we can see that more forgetting occurs in SMALL, even though trained for five times less number of global training steps. This phenomenon is further highlighted when comparing results from SMALL-P1 (8 epochs) which shows the most amount of forgetting despite being trained for ten times less number of global training steps. While the overall drop is much mitigated in Figure 3 (b) T5-Kadapters, we observe the same trend between each scenario which goes to show how critical observing the same data repeatedly during continued pretraining is for causing forgetting.

The results are in line with findings from Lee et al. (2021) which suggest LMs should be pretrained with just a few epochs on less duplicating data for efficiency. We add additional intuition to their findings and conjecture that the inefficiency of pretraining from duplicate data could have been caused by the forgetting of the rather long-tail knowledge in the pretraining corpus.

## 6 CONCLUSION

In this paper, we propose CONTINUAL KNOWLEDGE LEARNING (CKL), where we establish benchmark datasets and metrics, and explore methodologies towards continual knowledge learning of an ever-changing LM. We find that parameter-expansion methods show the most robust performance throughout all of the experimental settings, which nevertheless has severe memory inefficiency and that seeing the same data often is a critical cause of forgetting. We also discuss several other interesting results of which we leave further exploration to future studies. To this end, we suggest the community to explore CKL for the better design of an ever-changing LM.

ACKNOWLEDGMENTS

The authors would like to thank Sang-Woo Lee, Jinheon Baek, Miyoung Ko, Hyunji Lee, and Eunbi Choi for helpful discussions. This work was supported by Institute of Information & communications Technology Planning & Evaluation (IITP) grant funded by the Korea government (MSIT) (No. 2019-0-00075, Artificial Intelligence Graduate School Program (KAIST)).

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

# A  EXTENSION OF RELATED WORKS

As mentioned in Section 2, there are fundamental differences between the traditional CL formulations and CKL which make the previous CL methods inadequate for the CKL setting. In this section, we introduce the prior traditional continual learning methods in detail, explore the methods from the literature set as baselines for the CKL benchmark and how they address the identified limitations of CL methods, and provide descriptions about alternative methods making LMs cope with the changing world.

## A.1  TRADITIONAL CONTINUAL LEARNING

Traditional continual learning (CL) methods focus on addressing two aspects of transfer between sequentially incoming tasks: *forward transfer* and *backward transfer* (Lopez-Paz & Ranzato, 2017). *Forward transfer* refers to how past tasks affect the performance of the current and future tasks. *Backward transfer* refers to how current or future tasks affect the performance of previous tasks. The general pretrain-finetune approach can be seen as an instance of *positive forward transfer* where a model performs better on a target task after being pretrained on a more general source task. Moreover, catastrophic forgetting can be seen as an instance of *negative backward transfer* where previous tasks suffer performance due to continued training on different tasks. With respect to these two aspects, CL approaches can be categorized into three main approaches: regularization, rehearsal, and parameter-expansion methods.

**Regularization**  Elastic Weight Consolidation (EWC) (Kirkpatrick et al., 2017) is a method that regularizes important parameters of previous tasks while training for the current tasks, helping mitigate *the negative backward transfer* of previous tasks. Important parameters are measured via a Fisher information matrix computed by measuring the magnitude of the gradient update step of each parameter during training of previous tasks.

**Rehearsal**  Gradient Episodic Memory (GEM) (Lopez-Paz & Ranzato, 2017) is one of the first rehearsal methods that utilize samples from each task stored in *episodic memory* and places an inequality constraint with respect to the losses of the samples in order to prevent *negative backward transfer* as well as allow the *positive backward transfer*. Other methods such as Experience replay and local adaptation (d'Autume et al., 2019) replay samples stored in the memory of previous tasks during training to mitigate forgetting.

**Parameter-expansion**  Progressive Neural Networks (PNN) (Rusu et al., 2016) is one of the earliest parameter-expansion/sharing approaches that introduce new sets of parameters for each new task where previous parameters are frozen and can be connected via lateral connections allowing for *positive forward transfer*. PNN not only prevents *negative backward transfer* but also surpassed the previous pretrain-finetune approach in terms of *positive forward transfer* in some tasks.

## A.2  CKL METHODS FOR LANGUAGE MODELS

As mentioned in Section 2, we explore the methods from the literature that have addressed the limitations of CL methods and thus are applicable to CKL. We also categorize these methods into the three main categories of CL.

**Regularization**  Most CL methods that utilize regularization require computing important parameters of the previous task, which in this case is pretraining on the original text corpus. Determining these parameters is oftentimes unrealistic since it requires large-scale pretraining which can hardly be replicated by most. Also, exactly how and where the knowledge is stored in the parameters of an LM is currently extremely difficult to identify and localize (Vig et al., 2020; De Cao et al., 2021). RecAdam (Chen et al., 2020) overcomes this limitation by following the same training objective as EWC (Kirkpatrick et al., 2017) with a stronger independent assumption and places a quadratic penalty, ridding the need to access the initial pretraining corpus.

**Rehearsal**  Large LMs are usually pretrained on a vast amount of raw text corpus such as Common Crawl[9]. When treating pretraining as a CL task, limitations exist when trying to apply previous rehearsal methods since a few samples from the pretraining corpus cannot represent the overall world knowledge from the original pretraining corpus. Mix-Review (He et al., 2021) solves this issue by performing preliminary experiments in a smaller pretraining setting by assuming access to the pretraining corpus during finetuning and mixing random subsets of pretraining corpus depending on a mix-ratio that anneals towards the target task as training progresses. Mix-Review can be considered a mild version of multi-task learning.

**Parameter-expansion**  K-Adapter (Wang et al., 2021b) shares and freezes the original parameters and adds new parameters through adapters for continued pretraining of factual and linguistic knowledge and improve performance on three different knowledge-driven downstream tasks. More recently, LoRA (Hu et al., 2021) freezes the original parameters and injects trainable rank-decomposition matrices into each layer of the Transformer architecture, greatly reducing the number of trainable parameters and the computational hardware requirement while performing on-par or better than training all of the parameters. Both methods hypothesize freezing the original parameters allows mitigation of catastrophic forgetting. We test out the hypothesis through implementation in our CKL benchmark.

### A.3  METHODS OF INTEGRATING WORLD KNOWLEDGE WITH LANGUAGE MODELS

**Explicit Methods**  Facts-as-Experts (Verga et al., 2021) store representations of entities in the form of key-value pairs into external memory that can be modified during inference time. RAG (Lewis et al., 2020a) accesses a dense vector index of Wikipedia with a retriever and swaps indexes for updating the behavior of the model as the world changes. Blender Bot 2.0 (Xu et al., 2021; Komeili et al., 2021), is also one of the explicit methods that search the internet for recent knowledge and saves recent conversations in external long-term memory. Explicit methods, such as swapping indexes, adding explicit entity-relation knowledge, or searching the internet are in need of manual intervention during inference or are bound to tasks that require retrieval. In this paper, we focus only on implicit methods.

**Implicit Methods**  Zhu et al. (2020) proposed a new task of explicitly modifying specific facts without forgetting unmodified facts and provided several benchmark approaches without utilizing non-parametric memory, including constrained layer-wise finetuning. Wang et al. (2021b) proposed K-Adapter, a method that adds adapters to frozen layers of pretrained LMs to inject factual and linguistic knowledge and improve performance on downstream tasks. Chen et al. (2020) proposed a new optimizer that simulates the pretraining optimization while finetuning on the target task without needing access to the pretraining corpus, improving performance on the GLUE benchmark. De Cao et al. (2021) propose using a hyper-network to edit factual knowledge.

Even though these implicit methods are efficient methods of injecting or modifying knowledge from the implicit parameters of the LMs, they are all limited to injecting *specific knowledge* such as the case of (Wang et al., 2021b) or modifying *past knowledge* such as the case of (Zhu et al., 2020; De Cao et al., 2021). No work, to the best of our knowledge, has specifically addressed the *catastrophic forgetting* of world knowledge gained from the initial pretraining when continued pretraining on new text corpus for the gain of *new* world knowledge.

## B  DATASET CONSTRUCTION

In this section, we describe the dataset construction process we undergo in creating the benchmark datasets used in CKL. For the construction, we use Amazon Mechanical Turk (mturk)[10] for crowd-sourcing Human Intelligent Tasks (HITs) and separately hire 11 experts for annotation that requires extensive searching of the C4 corpus. In addition, three more experts[11] who set up the data construction process and prepared the annotation guideline to ensure the quality of the data through post-validation and giving feedback to the annotators in real-time. The interfaces used for mturk HITs are provided in Appendix B.2.

---

[9]https://commoncrawl.org/

[10]https://www.mturk.com

[11]The first three authors of the paper.

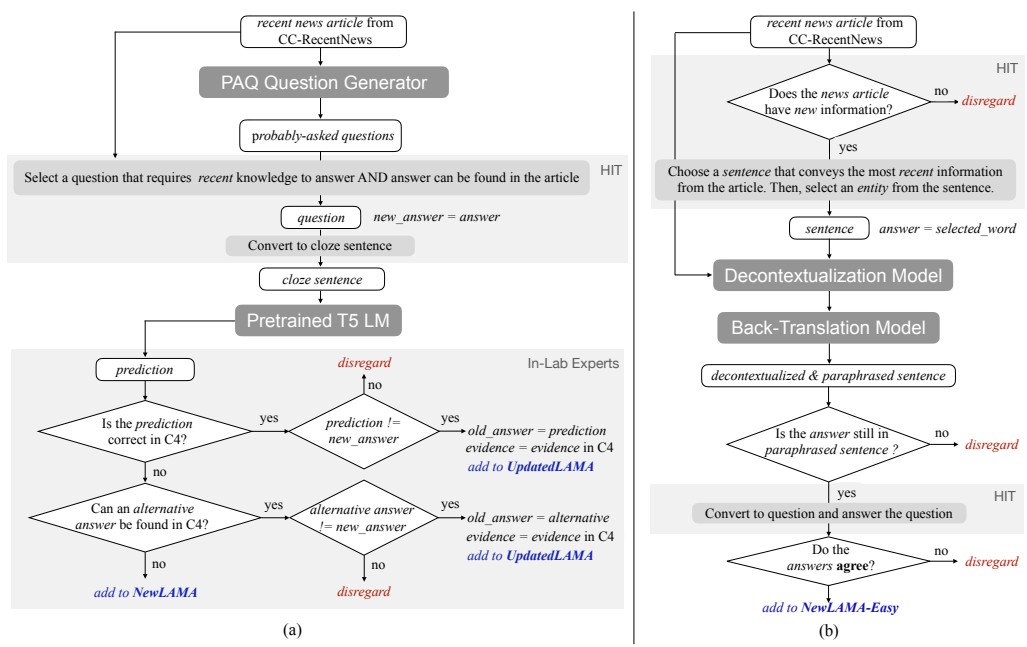

Figure 4: Dataset construction pipeline for (a) UPDATEDLAMA, NEWLAMA, and (b) NEWLAMA-EASY

**CC-RECENTNEWS** We first construct CC-RECENTNEWS, a novel text corpus containing relatively *new* knowledge as $D_1$. We use news-please (Hamborg et al., 2017), similar to the CC-NEWS (Liu et al., 2019) and REALNEWS dataset (Zellers et al., 2019), to crawl 221,779 news articles published from May 2020 to April 2021. LMs initially pretrained on $D_0$ constructed before May 2020 can be continually pretrained on CC-RECENTNEWS to gain relatively *recent* world knowledge.

**INVARIANTLAMA** We create INVARIANTLAMA, a subset of the LAMA (Petroni et al., 2019) task for measuring *time-invariant* knowledge which might be forgotten during CKL. Among the 41 relations of the T-REx (Elsahar et al., 2018) subset of LAMA, we manually select 28 relation types that probe for *time-invariant* instances (a full list of *time-invariant* relations are provided in Appendix B.1). We also remove instances where the answer overlapped with the subject following Poerner et al. (2019) since the answers for these instances can be inferred from the cloze statement itself. Lastly, we remove instances where the answer was a non-entity to leave only the instances that require world knowledge for prediction on their answers (Guu et al., 2020).

**UPDATEDLAMA and NEWLAMA** We construct UPDATEDLAMA and NEWLAMA for measuring the update of outdated knowledge and acquisition of new knowledge during CKL. The challenge of constructing UPDATEDLAMA is that a knowledge instance can be only considered as the knowledge that requires update only if it is present in both $D_0$ and $D_1$ with changed details, and the challenge of constructing NEWLAMA is that the knowledge can be considered new only if it is in $D_1$ but not in $D_0$. Therefore we set up the data construction process carefully. The pipeline for the creation of a single instance of UPDATEDLAMA and NEWLAMA, is shown in Figure 4 (a). Each potential instance starts off from a single article from CC-RECENTNEWS and goes through the pipeline which will end up being (1) discarded (2) added to UPDATEDLAMA or (3) added to NEWLAMA in the end. The procedure is as follows:

(1) First, a list of Probably-Asked Questions (Lewis et al., 2021) are generated using the PAQ question generator on a single news article from CC-RECENTNEWS. (2) The list of PAQs and the news article is given to the crowd-sourced worker to select a question that asks for the most *recent* knowledge for which the answer (denoted as *new answer*) can be found in the article. (3) The crowd-source worker is instructed to convert the question into a cloze sentence so that it can be given as input to a pretrained T5 LM. The predictions of the T5 LM are stored along with the questions and cloze sentences. (4) The expert annotator ensures the quality of the questions and cloze sentences by cor-

recting them whenever necessary and checks whether the model prediction is correct by searching through the C4 corpus as a representative of $D_0$[12]. If the prediction is correct and the prediction is not the same with the *new answer*, the following instance must be present in both $D_0$ and $D_1$ with details changed, and thus is added to UPDATEDLAMA along with the evidence document found in C4. If same, the instance is discarded because the instance is neither *updated* nor *new*. (5) Lastly, if the model prediction is wrong, the expert annotator is asked to find an alternative answer for the question in C4. If not found, the instance is added to NEWLAMA since the answer to the question could only be found in the article of CC-RECENTNEWS ($D_1$), but not in C4 ($D_0$). Similarly, if the alternative answer is found in C4, we check whether it is the same as the *new answer* and add the instance to UPDATEDLAMA if not the same and disregard it otherwise.

Throughout the whole process, a validator checks the sanity of the data and gives detailed real-time feedback on the work of the annotator.

**NEWLAMA-EASY** Even though NEWLAMA corresponds to our exact definition of *new knowledge* that we define in the task formulation, scaling the size of the dataset was difficult since each instance required searching the whole C4 database for answers. Instead, we provide a much larger, *easier* variant NEWLAMA-EASY where we test the general new knowledge acquired during continued pretraining on CC-RECENTNEWS. The pipeline for the creation of a single instance of NEWLAMA-EASY is shown in Figure 4 (b) and follows the following procedures:

(1) First, the crowd-sourced worker is instructed to classify whether the given article contains *new* information or not. (We define *new* as not likely to be known before May 2020). If the article contains new information, the worker is instructed to select a sentence from the article that contains the most *recent* information and an *entity* among the possible answer candidates in the sentence and discard the article if otherwise. We provide the possible entities through a Named-Entity Recognition Model. (2) We make the selected sentence *stand-alone* from the article through the decontextualization model provided by Choi et al. (2021). (3) The decontextualized sentence is paraphrased by a back-translation model (en→de→en) (Tiedemann & Thottingal, 2020) and checked whether the selected word is still in the paraphrased sentence; the sentence is discarded if not. (4) Next, we mask out the selected word from the sentence and ask two crowd-sourced workers to convert the cloze sentence into a question and answer the question. (5) If the answers agree among the workers as well as correspond to the actual selected word, we add the instance to NEWLAMA-EASY.

The specific interfaces used for the mturk HITs are provided in Appendix B.2. Statistics of the constructed datasets are in Appendix B.3.

### B.1 TIME-INVARIANT RELATIONS OF LAMA

Table 4 shows the list of 28 time-invariant relations of INVARIANTLAMA. We manually filter the 44 original LAMA relations to leave only the time-invariant relations. Templates such as "[X] works for [Y] ." and "[X] is a member of [Y] ." are excluded because the answer may change for different timestamps. In the template, [X] and [Y] refers to subject and object labels, respectively. Given a template with only the subject included, the model has to predict the object label [Y] for knowledge probing.

### B.2 INTERFACES USED FOR THE CONSTRUCTION OF CKL BENCHMARK

The Mturk interface used during construction of UPDATEDLAMA and NEWLAMA, NEWLAMA-EASY, and NEWLAMA-EASY are shown in Figure 5, 6, and 7, respectively.

### B.3 DATASET STATISTICS AND EXAMPLES

We report the data statistics for the CKL benchmark in Table 5. We measure the size, average input token length, average answer token length, and the answer types of each constructed dataset. One thing to consider is that LAMA (Petroni et al., 2019) from which we constructed INVARIANTLAMA is originally constructed for only single-token decoding (1.3 with the T5-tokenizer) because multi-token decoding entails additional, tunable parameters (beam size, n-gram repetition penalties, etc.).

---

[12]The expert annotators are instructed to use https://c4-search.apps.allenai.org/ for searching through the C4 corpus.

Table 4: Relations of INVARIANTLAMA

| Relation | Template ([X], [Y]) | Example |
|---|---|---|
| P19 | [X] was born in [Y] . | Taras Kuzio was born in Halifax . |
| P20 | [X] died in [Y] . | Georgios Roilos died in Athens. |
| P279 | [X] is a subclass of [Y]. | Hutterite German is a subclass of Bavarian . |
| P37 | The official language of [X] is [Y]. | The official language of Azad Kashmir is English . |
| P449 | [X] was originally aired on [Y] . | Microsoap was originally aired on BBC. |
| P47 | [X] shares border with [Y] . | Illinois shares border with Kentucky . |
| P138 | [X] is named after [Y] . | Logan International Airport is named after Boston . |
| P364 | The original language of [X] is [Y] . | The original language of The Fatal Eggs is Russian . |
| P527 | [X] consists of [Y] . | AIM alliance consists of Apple . |
| P176 | [X] is produced by [Y] . | Alfa Romeo 155 is produced by Fiat . |
| P27 | [X] is [Y] citizen . | Woodrow Lloyd is Canada citizen . |
| P407 | [X] was written in [Y] . | France Culture was written in French . |
| P30 | [X] is located in [Y] . | Lavoisier Island is located in Antarctica . |
| P178 | [X] is developed by [Y]. | Tizen is developed by Intel . |
| P1376 | [X] is the capital of [Y], | London is the capital of England . |
| P131 | [X] is located in [Y] . | Pershing County is located in Nevada . |
| P1412 | [X] used to communicate in [Y]. | Jacques Rivette used to communicate in French . |
| P17 | [X] is located in [Y] . | Eibenstock is located in Germany . |
| P276 | [X] is located in [Y] . | Delhi Technological University is located in India . |
| P937 | [X] used to work in [Y]. | Pierre Trudeau used to work in Ottawa . |
| P140 | [X] is affiliated with the [Y] religion . | Emirate of Granada is affiliated with the Islam religion . |
| P103 | The native language of [X] is [Y] . | The native language of Anastasy Vonsyatsky is Russian . |
| P190 | [X] and [Y] are twin cities . | Beijing and Milan are twin cities . |
| P1001 | [X] is a legal term in [Y] . | Surgeon General is a legal term in Canada . |
| P495 | [X] was created in [Y] . | La Grande Vadrouille was created in France . |
| P36 | The capital of [X] is [Y] . | The capital of Granville County is Oxford . |
| P740 | [X] was founded in [Y]. | Grimaldi Group was founded in Naples . |
| P361 | [X] is part of [Y] . | Sinqa is part of Andes . |

Table 5: CKL benchmark dataset statistics

| Dataset | Size | Avg. Input Token # | Avg. Answer Token # | Answer Types |
|---|---|---|---|---|
| INVARIANTLAMA | 17474 | 11.9 | 1.3 | Geographical (54%), Language (14.9%), Nationalities (7.2%) Person (6.3%), Location (5.7%), Organization (5.3%), etc. (6.6%) |
| UPDATEDLAMA | 924 | 13.7 | 9.4 | Person (61.47%), Organization (8.3%), Geographical (6.6%), Numerals (5.19%), Date (2.4%), etc. (16.04%) |
| NEWLAMA | 797 | 14.7 | 8.7 | Person (59.7%), Organization (10.2%), Numerals (7.6%) Date (5.3%), Geographical (4.8%), etc. (12.4%) |
| NEWLAMA-EASY | 11177 | 44.4 | 6.1 | Person (48.5%), Organization (13%), Geographical (9.8%) Date (5.5%), Nationalities (3.4%), Numerals (2.5%), etc. (17.3%) |

The newly constructed datasets UPDATEDLAMA, NEWLAMA, and NEWLAMA-EASY require multi-token decoding which adds a level of difficulty for the task compared to INVARIANTLAMA. Moreover, NEWLAMA-EASY has a different input distribution (longer input sequences) than the other datasets since the decontextualization and back-translation processes are applied to create each instance, which makes the sentences longer. Lastly, some examples of the CKL benchmark datasets are provided in Table 6.

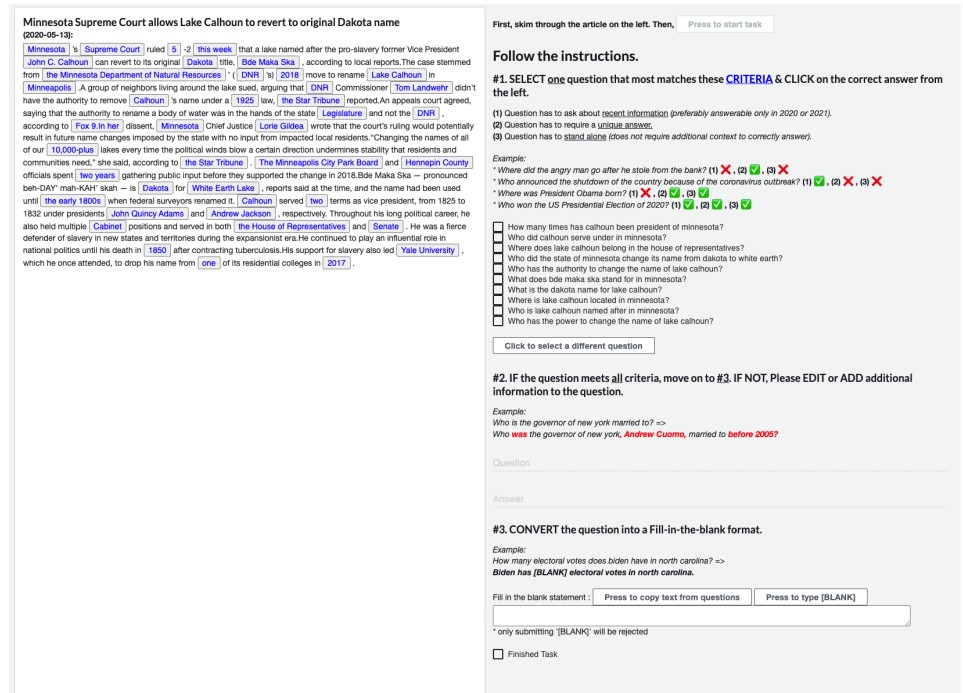

Figure 5: Mturk interface used for construction of UPDATEDLAMA and NEWLAMA

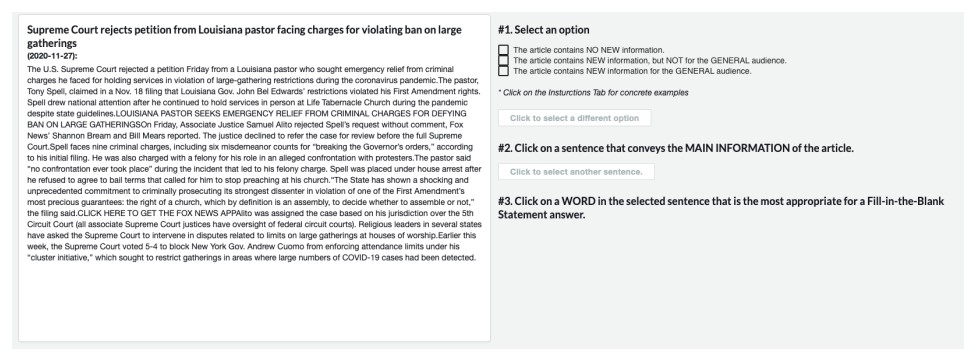

Figure 6: First mturk interface used for construction of NEWLAMA-EASY

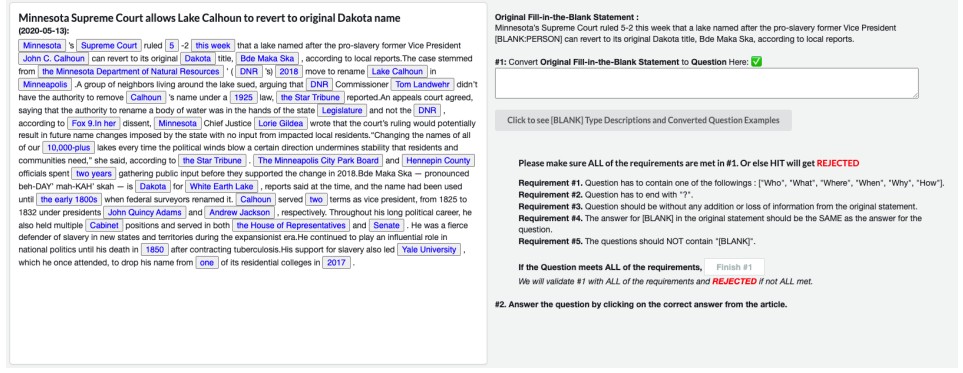

Figure 7: Second mturk interface used for construction of NEWLAMA-EASY

Table 6: Examples of INVARIANTLAMA, UPDATEDLAMA, NEWLAMA, and NEWLAMA-EASY

| Task | Input | Output |
|---|---|---|
| INVARIANTLAMA | iPod Touch is produced by _______. 
 The Sharon Cuneta Show was created in _______. 
 The native language of Lee Chang-dong is _______. | Apple 
 Philippines 
 Korean |
| UPDATEDLAMA | _______ is the prime minister of England. 

 _______ has the most passing yards in the NFL. 

 Bale has _______ champions league titles with Real Madrid. | Theresa May→ 
 Boris Johnson 
 Brady Quinn→ 
 Jalen Guyton 

 3→4 |
| NEWLAMA | Alicia Braga plays _______ in the New Mutant. 
 _______ owns the rights to the Falcon and the Winter Soldier. 
 Tesla invested _______ in the digital currency bitcoin. | Cecilia Reyes 

 Disney 

 1.5 billion |
| NEWLAMA-EASY | The decision of the two volleyball stars Bria and Cimone Woodard to withdraw from the Power 5 School to study at _______ has become a national story. 
 Allen Lazard is officially listed as questionable with a nuclear injury after missing the last _______ games. | Howard University 


 six |

## C  EXPERIMENTAL CONFIGURATION

**Pretraining Congifuration**   We utilize the T5 initially pretrained on C4 (April 2019) and continually pretrained with salient span masking (Guu et al., 2020) on Wikipedia (May 2020) as initialization. We use the checkpoints from Wolf et al. (2020). We also perform the SSM objective during CKL because it was shown to help LMs "focus on problems that require world knowledge" (Guu et al., 2020; Roberts et al., 2020).

**Continual Pretraining Configurations**   The input and output sequence length is fixed to 350. We use gradient accumulation for cases where the same number of training batches could not be loaded on the GPUs due to the varying memory consumption required for different methods and set the global batch size to 60. We use Adafactor optimizer with an initial learning rate of 1e-3. We show the effects of learning rate variation regarding the trade-off between maintaining previous knowledge and acquiring new knowledge in Appendix E. We use learning rate warm-up for the first 10% of training and linearly decay the learning rate to half of the initial learning rate towards the end of training. For all of the experiments, we use 4 32GB V100 GPUs for training with each method except Mix-Review, where we use 16 32GB V100 GPUs. The details of the configurations used for evaluation on each individual CKL task are provided in Appendix C.

**Evaluation Configurations**   For T5 based models, all evaluation is done in a zero-shot manner and is processed with a single GPU. For INVARIANTLAMA, the input and output length is fixed as 25 and 4 respectively. For UPDATEDLAMA and NEWLAMA, the input and output length is 50 and 10 respectively. Lastly, the input and output length is 150 and 10 respectively for NEWLAMA-EASY. The rationale of this hyperparameter is based on average input and answer token in Table 5.

Unlike T5 models, GPT-2 based models need additional *light-tuning* for 1 epoch for evaluation. For INVARIANTLAMA, the input and output length is 50 and 3 respectively. The training batch size is 32 and the learning rate is 1e-3. For evaluation on the acquisition of new knowledge, the input and output length is 100 and 10 respectively. The training batch size is 8 due to memory constraints and the learning rate is 1e-3. For both tuning processes, 4 V100 32GB GPUs are used. The detailed result and discussion of GPT-2 based models are shown in Appendix G.

## D  HYPERPARAMETERS FOR IMPLEMENTATION OF CKL METHODS

**RecAdam** (Chen et al., 2020) We use the same hyperparameter setting for the optimizer as in Chen et al. (2020): we set the coefficient of the quadratic penalty $\gamma$ to 5,000, and select the best $t_0$ and $k$ in 100, 250, 500, 1,000 and 0.05, 0.1, 0.2, 0.5, 1 respectively for the annealing coefficient $\lambda(t)$.

Table 7: Result of T5-Vanilla and T5-Kadapters continually pretrained with various learning rates. The experiments are done under the setting of SMALL scenario in Table 3, thus $D_0$ are C4 (April 2019) and Wikipedia (May 2020), and $D_1$ is CC-RECENTNEWS-SMALL. Each of IL and NLE stands for INVARIANTLAMA and NEWLAMA-EASY. The parameters of FUAR are $\mathbb{T}^F$, $T_1^U$, and $T_1^A$, the tasks measuring the amount of time-invariant knowledge from corpus $D_0$, updated knowledge from $D_1$, and newly acquired knowledge from $D_1$, respectively.

| Method | Learning Rate | IL
EM | NLE
EM | FUAR
$((\mathbf{IL}), \boldsymbol{n.d.}, \mathbf{NLE}) \downarrow$ |
|---|---|---|---|---|
| T5-Initial | - | **24.17** | 8.9 | - |
| T5-Vanilla | 1e-05 | 19.15 | 13.56 | 1.08 |
| T5-Vanilla | 1e-04 | 17.45 | 15.21 | 1.06 |
| T5-Vanilla | 5e-04 | 14.88 | 15.89 | 1.33 |
| T5-Vanilla | 1e-03 | 11.19 | 18.77 | 1.32 |
| T5-Kadapters (k=2) | 1e-04 | 19.93 | 14.93 | **0.70** |
| T5-Kadapters (k=2) | 1e-03 | 16.46 | **19.59** | 0.72 |

**Mix-Review** (He et al., 2021) We use the English Wikipedia [13] to represent the original pretraining corpus. The mix-decay and mix-ratio are set to 4 and 0.7, respectively, which is the best hyperparameter setting in the paper.

**LoRA** (Hu et al., 2021) We only freeze the encoder for the encoder-decoder LM and the entire model for the decoder-only LM. We use the optimal rank $r$ of 4 and adapt both $W_q$ and $W_v$ in the self-attention module, which corresponds to the best performing hyperparameter setting in the paper.

**K-Adapter** (Wang et al., 2021b) Similarly with T5-LoRA, we freeze the encoder for the encoder-decoder LM and the entire model for GPT-2. We implement $k = 2, 3$ for both T5 and GPT-2 to see the effect of increasing # of parameters. Unlike in the original paper, we set the configuration of the adapter identical to a single transformer layer from the original LM, ridding the need of an up-projection and down-projection layer.

**Modular** We use a projection layer before adding the hidden state outputs from both encoders to match the dimensions.

**Why do we add parameters to only the encoder for T5?** For parameter-expansion methods, we add parameters to only the encoder because the encoder is applied to the input sequence and the decoder is applied to the output sequence. Since most of the computational cost comes from the decoder computing for the output sequence in an auto-regressive manner as highlighted in (Li et al., 2021), the newly added parameters in the encoder are roughly expected to have minimal additional computational cost.

**Why do we freeze parameters of only the encoder for T5?** K-Adapter and LoRA are initially proposed to freeze all of the parameters except for the newly added parameters. However, when applying this methodology to T5, it was empirically shown that unfreezing the parameters of the decoder results in better performances when utilized together with parameter-expansion methods in terms of overall trade-off.

## E    EXPLORING THE TRADE-OFF OF VARYING THE LEARNING RATE FOR CONTINUAL PRETRAINING

Table 7 shows that lowering the learning rate for the continual pretraining leads to less forgetting of the original knowledge, but also less learning of new knowledge. The experiments are done under the setting of SMALL scenario in Table 3.

By comparing the FUAR among the T5-Vanilla models with different learning rates, it can be seen that there is no rule of thumb for choosing the appropriate learning rate since FUAR is the lowest

---

[13]https://huggingface.co/datasets/wikipedia

Table 8: Dev performance on KILT benchmark datasets after finetuning. Each model is finetuned on the train sets of KILT after continually trained on CC-RECENTNEWS dataset for 4 epochs.

| | Fact Checking | Entity Linking | | | Slot-filling | | Open Domain QA | | | | Dialogue |
|---|---|---|---|---|---|---|---|---|---|---|---|
| **Method** | **FEVER** | **AY2** | **WnWi** | **WnCw** | **T-REx** | **zsRE** | **NQ** | **HoPo** | **TQA** | **ELI5** | **WoW** |
| | ACC | ACC | ACC | ACC | ACC | ACC | EM | EM | EM | Rouge | F1 |
| T5-Initial | 80.39 | 81.44 | **50.47** | **48.92** | 44.64 | **4.40** | 25.63 | 17.64 | **28.38** | 13.46 | 13.92 |
| T5-Vanilla | 78.02 | 81.19 | 48.17 | 46.46 | 44.08 | 2.04 | 24.93 | 14.36 | 26.51 | 13.38 | 13.07 |
| T5-RecAdam | 77.83 | 81.44 | 49.12 | 47.01 | 43.04 | 2.58 | 24.65 | 14.86 | 25.99 | 13.71 | 12.69 |
| T5-MixReview | 77.17 | 80.77 | 49.38 | 46.22 | 44.08 | 2.47 | 25.07 | 14.57 | 26.36 | 13.57 | 12.73 |
| T5-LoRA | 79.89 | 81.44 | 48.82 | 47.29 | 45.68 | 3.01 | 25.49 | 16.71 | 28.23 | 13.42 | 13.60 |
| T5-Kadapters (k=2) | 80.35 | 80.94 | 48.91 | 46.65 | 45.52 | 3.33 | **26.20** | 16.57 | 26.89 | 13.15 | 12.94 |
| T5-Kadapters (k=3) | 80.31 | 80.52 | 47.09 | 46.26 | 45.60 | 3.12 | 24.79 | 16.57 | 25.62 | **13.82** | 13.42 |
| T5-Modular | **80.54** | **82.44** | 48.44 | 44.81 | **48.16** | 3.44 | 24.51 | **18.43** | 28.31 | 13.72 | **14.03** |

Table 9: Hyperparameters and dataset details for all tasks of KILT.

| | Fact Checking | Entity Linking | | | Slot-filling | | Open Domain QA | | | | Dialogue |
|---|---|---|---|---|---|---|---|---|---|---|---|
| | **FEV** | **AY2** | **WnWi** | **WnCw** | **T-REx** | **zsRE** | **NQ** | **HoPo** | **TQA** | **ELI5** | **WoW** |
| Epoch | 5 | 20 | - | - | 9 | 30 | 45 | 12 | 50 | 6 | 8 |
| Input Seq | 25 | 768 | 512 | 2,048 | 25 | 25 | 35 | 50 | 25 | 35 | 175 |
| Output Seq | 10 | 6 | 6 | 6 | 6 | 6 | 6 | 8 | 10 | 350 | 40 |
| LR | 1e-4 | 1e-4 | - | - | 1e-3 | 1e-4 | 1e-3 | 1e-4 | 1e-3 | 1e-3 | 1e-4 |
| Batch Size | 128 | 16 | 128 | 48 | 512 | 256 | 256 | 256 | 128 | 32 | 64 |
| Train Size | 104,966 | 18,395 | - | - | 2,284,168 | 147,909 | 87,372 | 88,869 | 61,844 | 272,634 | 63,734 |
| Dev Size | 10,444 | 4,784 | 3,396 | 5,599 | 5,000 | 3,724 | 2,837 | 5,600 | 5,359 | 1,507 | 3,054 |

in learning rate of 1e-4 and increases for both lower and higher learning rates. We suppose that the optimal learning rate heavily depends on the corpus size of $D_1$ and the model capacity of LM. We also report the performance of T5-Kadapters, which is a CKL method that shows robust performance throughout most experiments. Applying T5-Kadapters consistently mitigates the trade-off between forgetting and acquiring new knowledge as shown by the improvement in FUAR from the T5-Vanilla model with the same learning rates, although the level of effectiveness varies according to the value of the learning rate. We do not perform extensive experiments with each of the varying learning rates since searching for the optimal learning rate for each different continued pretraining setting may be out-of-scope with this research.

## F  EXPLORING HOW CONTINUALLY PRETRAINING ON $D_1$ AFFECTS KILT TASKS WHICH REQUIRES KNOWLEDGE FROM $D_0$

In addition to the CKL benchmark, we also show in Table 8 the performance on the dev set of KILT (Petroni et al., 2021) after finetuning each of the continually pretrained models of Table 2. Since KILT is made from Wikipedia, which corresponds to the old pretraining corpus $D_0$, the performance on KILT measures how continual pretraining on new corpus $D_1$ affects the performance on the knowledge obtained from $D_0$ if finetuning is done on behalf of the knowledge from $D_0$.

**Configuration**  KILT (Petroni et al., 2021) consists of 5 different tasks and 11 datasets: Open-Domain Question Answering (Joshi et al., 2017; Kwiatkowski et al., 2019; Fan et al., 2019; Yang et al., 2018), Fact Checking (Thorne et al., 2018), Entity Linking (Hoffart et al., 2011; Guo & Barbosa, 2018), Slot-filling (Levy et al., 2017), and Knowledgeable Open Dialogue (Dinan et al., 2019). Because each task requires a different training objective than the one used during pretraining, additional finetuning is necessary. We search for the hyperparameters such as training epochs, batch size, input size, output size, and learning rate of each individual KILT task to match the T5-base dev performance reported by Petroni et al. (2021). Using the identified configurations, we perform experiments on all of the KILT tasks with the continually pretrained models for each method as the initialization checkpoints. Evaluation metrics are different for each dataset: accuracy for discrete

output (fact-checking, entity linking, slot-filling), Exact Match (EM) for question answering tasks with short output, ROUGE-L for ELI5 (question answering task with long output), and F1-score for Wizard of Wikipedia (dialogue). The data statistics and the hyperparameters used for finetuning on each KILT dataset is reported in Table 9.

**Experimental Result** We first focus on the performance on zero-shot Relation Extraction (zsRE), which is measured on the dev set of 12 relations that are ensured to have no overlap with the 84 relations of the train set (Levy et al., 2017). Since the setting is similar to the zero-shot probing setting of IL, the trend of the result on the two datasets are similar. The performance of T5-Vanilla drops to half from that of T5-Initial as shown in IL, and the best performing method for both datasets is T5-Modular. In addition, corresponding with results from the CKL benchmark, parameter-expansion methods generally show stronger performance than the other methods.

However, for the other datasets that cannot be performed in a zero-shot manner, the intermediate process of continually pretraining on corpus $D_1$ does not seem to be that harmful on the finetuning for the target tasks even though they are more related to the knowledge of $D_0$. Even T5-Vanilla shows modest performance, sometimes with better results than some other CKL baselines. One hypothesis is that the models could have regained the original knowledge from corpus $D_0$ through the finetuning process. Also, some of the knowledge could have been recovered through the test-train overlap (Lewis et al., 2020b; Wang et al., 2021a).

A more surprising finding is that the performance of some of the parameter-expansion methods are even higher than that of T5-Initial, which is considered to be the upper bound for KILT because T5-Initial is only trained on behalf of the knowledge from $D_0$. For example, T5-Modular shows higher scores than T5-Initial on 6 out of 11 tasks. Since the parameter-expansion methods force the model to store the new knowledge in the newly added parameters during continual pretraining, one careful conjecture is these LMs have learned to combine and utilize in its internal representation of both old and new knowledge stored in separate parameters during finetuning to maximize the performance.

## G    EXPLORING HOW CKL METHODS TRANSFER ACROSS LM ARCHITECTURES

We perform experiments with GPT-2 Large (∼ 774M params) (Radford et al., 2019) initially pretrained on WebText and Wikipedia[14] ($D_0$) and continually trained on CC-RECENTNEWS-SMALL, i.e., SMALL ($D_1$) for 8 epochs. For continued pretraining, we use the common teacher-forcing pretraining objective. The initial learning rate for the continued pretraining stage is empirically chosen as 1e-4 (results with learning rate as 1e-3 are shown in Appendix G.1). After continued pretraining, we apply *light-tuning*, a process denoted for finetuning the model for only one epoch on a small portion of data similar to the evaluation set. Training on a single epoch constrains the model to barely adapt to the input-output form of the data and not to learn the knowledge in tuning samples, mitigating the problem suggested by Lewis et al. (2020b).

To measure the time-invariant knowledge, we use InvariantLAMA (IL) because most of the slots to fill are at the end of the sentence. For light-tuning on behalf of IL, we use additional T-Rex data from Shin et al. (2020) which has a similar distribution as instances from IL. Among them, 5,000 instances with the same *time-invariant* relations as IL are randomly sampled for *light-tuning*. On the other hand, unlike IL where most of the slots to fill are at the end of the sentences, the LAMA datasets for new knowledge in our CKL benchmark mostly have the slots at the beginning of the sentences. Therefore, we use the corresponding CBQA dataset of NEWLAMA-EASY, NEWQUESTIONS-EASY (NQE) to roughly measure the new knowledge.[15] For light-tuning on behalf of NQE, 5,000 instances are sampled from a set of QA pairs constructed from CC-RECENTNEWS but not CC-RECENTNEWS-SMALL to remove the test-train overlap.

---

[14]GPT-2 was initially pretrained on WebText (Dec 2019), which consists of 8 million documents with Wikipedia pages excluded. In order to measure the performance on INVARIANTLAMA constructed from Wikipedia, we continually pretrain GPT-2 on a subset of Wikipedia (May 2020) for 14k global training steps before CKL.

[15]The QA version of UL, NL and NLE will be also released with the main CKL benchmark.

Table 10: Performance of decoder-only models initially pretrained on Dec 2019 dump of Webtext and May 2020 dump of Wikipedia ($D_0$) continually pretrained on CC-RECENTNEWS-SMALL ($D_1$) for 8 epochs with a learning rate of 1e-4. Each of IL and NQE stands for INVARIANTLAMA and NEWQUESTIONS-EASY. The parameters of FUAR are $\mathbb{T}^F$, $T_1^U$, and $T_1^A$, the tasks measuring the amount of time-invariant knowledge from corpus $D_0$, updated knowledge from $D_1$, and newly acquired knowledge from $D_1$, respectively.

| Method | IL EM | NQE EM | FUAR $((\mathbf{IL}), \boldsymbol{n.d.}, \mathbf{NQE}) \downarrow$ |
|---|---|---|---|
| GPT2-Initial | 38.11 | 4.3 | - |
| GPT2-Vanilla | 35.88 | 5.79 | 1.58 |
| GPT2-Recadam | 35.50 | 5.79 | 1.84 |
| GPT2-Mixreview | **38.93** | 5.57 | **0** |
| GPT2-Lora | 37.99 | 6.23 | 0.06 |
| GPT2-Kadapters (k=2) | 37.85 | **6.34** | 0.13 |
| GPT2-Kadapters (k=3) | 38.03 | 5.79 | 0.06 |

Table 10 shows the CKL benchmark performance of GPT-2 models. We report the results averaged over 5 runs with different random seeds. As in Table 2, parameter-expansion methods show robust performance on both IL and NQE, resulting in low FUAR. This shows that these methods are not only effective on the encoder-decoder model but also the decoder-only model as well. One interesting result in Table 10 is that GPT2-MixReview performs the best on IL, with performance even higher than the initial model, which results in the best FUAR of 0 which means no forgetting occurred at all. We suppose that the training strategy of GPT2-MixReview, allowing access to samples of $D_0$ during continued pretraining, would have allowed fast adaptation to knowledge from $D_0$ during the *light-tuning* phase. Performance of GPT2-MixReview suggests that it makes it possible to regain the original knowledge for decoder-only models even with small tuning steps.

We want to highlight that the discrepancy of the performances among the CKL methods between encoder-decoder LM (T5) and decoder-only LM (GPT-2) may not solely be on the LM architecture, but also on the learning rate and the evaluation method (light-tuning was used to evaluate GPT-2 while we evaluated T5 in a zero-shot manner). We leave further exploration of training ever-changing decoder-only LMs such as GPT-2 as future work.

### G.1 FAILED GPT-2 EXPERIMENTS WITH LARGER LEARNING RATE

Table 11 shows the CKL benchmark result of GPT-2 models continually pretrained on CC-RECENTNEWS-SMALL for 8 epochs with a learning rate of 1e-3. By comparing the results in this table with those in Table 10, which is for models continually pretrained with a learning rate of 1e-4, the results in Table 11 shows worse performance on both IL and NQE. Unlike in Appendix E, increasing the learning rate does not result in better learning of new knowledge. Instead, NQE performance is even worse than GPT2-Initial for GPT2-Vanilla, GPT2-Recadam, and GPT2-MixReview. FUAR is *no gain* for these cases by the definition of the metric because the denominator has the value of zero. This shows that a large learning rate for continual pretraining may lead to failure: neither retaining old knowledge nor acquiring new knowledge effectively. For parameter-expansion methods, because many parameters including the decoder are frozen during the continual training process, they seem to be less prone to the effect of a large learning rate.

## H EXPLORING THE PREDICTION CHANGE DURING CONTINUAL PRETRAINING

Table 12 shows the prediction results of T5-Vanilla and T5-Modular on three knowledge probing tasks: INVARIANTLAMA, UPDATEDLAMA, and NEWLAMA. We show the prediction for every training epoch for each model. The instances are selected from the predictions that T5-Modular got correct but T5-Initial got wrong on the final prediction, in order to see where the gap of the EM comes from.

Table 11: Performance of decoder-only models initially pretrained on Dec 2019 dump of Webtext and May 2020 dump of Wikipedia ($D_0$) continually pretrained on CC-RECENTNEWS-SMALL ($D_1$) for 8 epochs with a learning rate of 1e-3. These are the results failed due to a large learning rate. Each of IL and NQE stands for INVARIANTLAMA and NEWQUESTIONS-EASY.

| Method | IL | NQE | FUAR |
| | EM | EM | $((\mathbf{IL}), \boldsymbol{n.d.}, \mathbf{NQE}) \downarrow$ |
| --- | --- | --- | --- |
| GPT2-Initial | **38.11** | 4.37 | - |
| GPT2-Vanilla | 23.03 | 1.64 | *no gain* |
| GPT2-Recadam | 25.38 | 2.73 | *no gain* |
| GPT2-Mixreview | 32.07 | 1.64 | *no gain* |
| GPT2-Lora | 34.52 | 5.46 | 3.29 |
| GPT2-Kadapters (k=2) | 33.67 | 6.01 | 2.71 |
| GPT2-Kadapters (k=3) | 31.75 | **7.65** | **1.94** |

Table 12: Change of Prediction Outputs During Continued Pretraininig

| | Cloze Sentence | Model | Epoch 1 | Epoch 2 | Epoch 3 | Epoch 4 | Answer |
| --- | --- | --- | --- | --- | --- | --- | --- |
| **IL** | The native language of Yvonne Monlaur is ___ . | V
M | French
French | French
French | Khmer
French | Malaya
French | French |
| | Sonic Drift 2 is developed by ___ . | V
M | Sonic D
Sonic R | Sonic the
Sega | Sonic Found
Sega | Sonic the
Sega | Sega |
| | WebKit is developed by ___ . | V
M | Microsoft
Apple | Google
Apple | GitHub
Apple | Google
Apple | Apple |
| | The official language of Republic of Ingushetia is ___ . | V
M | Russian
Russian | English
Russian | Kazakh
Russian | English
Russian | Russian |
| | The capital of Roman Empire is ___ . | V
M | Rome
Rome | Rome
Rome | Constantino
Rome | Constantino
Rome | Rome |
| **UL** | The biggest exporter of crude oil to china is ___ . | V
M | Saudi Arabia
Russia | Saudi Arabia
Saudi Arabia | Saudi Arabia
Russia | Saudi Arabia
Russia | Saudi Arabia → Russia |
| | ___ is the head of the euro zone central bank | V
M | Mario Draghi
Mario Draghi | Yves Le Maire
Christine Lagarde | Yves Dujarric
Christine Lagarde | Mario Draghi
Christine Lagarde | Mario Draghi → Christine Lagarde |
| | ___ is the manager of chelsea in the premier league | V
M | Mauricio Fernandez
Jose Mourinho | Steve Bruce
Jose Mourinho | Frank Lampard
Frank Lampard | Mikel Arteta
Frank Lampard | Luis Enrique → Frank Lampard |
| | ___ is the price for a flat in nottingham | V
M | What
This | 999
30,000 pounds | £1.25m
40,000 pounds | £1.25m
40,000 | 36,000 → 40,000 |
| | ___ was the governor of New York at the time this article was written | V
M | Andrew M. Cuomo
Andrew Cuomo | Cuomo
Andrew Cuomo | Andrew Cuomo
Andrew M. Cuomo | Franklin D. Roosevelt
Andrew Cuomo | Martin Van Buren → Andrew Cuomo |
| **NL** | ___ is on the Bills all-pro team | V
M | Corey
Williams | Williams
Williams | Corey
Williams | Connor
Williams | Williams |
| | ___ is the founder of the popular cryptocurrency bitcoin | V
M | Satoshi Nakamoto
Vitalik Buterin | Satoshi Nakamoto
Satoshi Nakamoto | Yuri
Satoshi Nakamoto | Xiaobo
Satoshi Nakamoto | Satoshi Nakamoto |
| | The bail for kyle rittenhouse is ___ . | V
M | Rs. 1 crore
$2 million | a whopping $1 million
$2 million | $2 million
$2 million | $1 million
$2 million | $2 million |
| | The las vegas raiders beat ___ in the playoffs | V
M | the Las Vegas Raiders
the New Orleans Saints | the New Orleans Saints
the Kansas City Chiefs | the Las Vegas Raiders
the Kansas City Chiefs | the sacramento
the New Orleans Saints | the New Orleans Saints |
| | ___ is the host of ellen de generes show | V
M | Yves
Elise | samantha s
Ellen DeGeneres | Norma
Ellen deGenes | Mike
Ellen DeGeneres | Ellen DeGeneres |

# I  EXPLORING THE CAUSE OF THE EM GAP BETWEEN UPDATEDLAMA AND NEWLAMA

As shown in the main experiment, Table 2, there is a considerable gap between the EM of UP-DATEDLAMA (UL) and NEWLAMA (NL) over all the methods, despite undergoing the same data construction process. We attempt to analyze the causation by first analyzing what answer types make up the EM score of both UL and NL of T5-Vanilla, which are 10.17 and 3.77, respectively. As shown in Figure 8a, the cloze sentences that take *Person* type as the ground truth makes up most of the EM of both tasks, despite *Person* type answers taking up a similar proportion out of the total answer types (61.46% for UL and 59.7% for NL). Since UL consists of probes requiring an update of information from $D_0$, one might conjecture that the EM gap is simply due to the difference of the

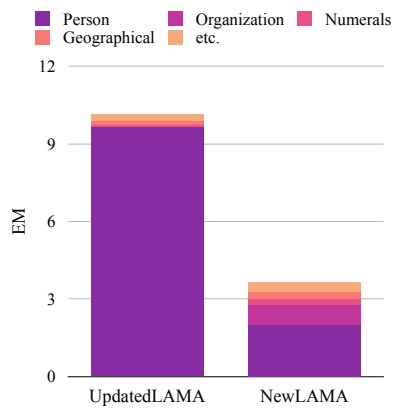

(a) Composition of ground truth categories of the correctly predicted instances

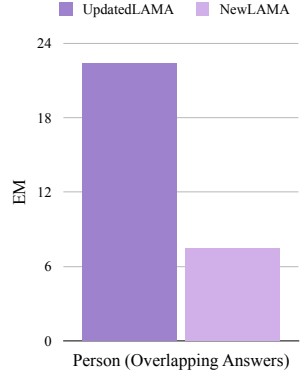

(b) EM measured using the instances from UL and NL with overlapping *Person* type answers

Figure 8: Analyzing the cause of the EM gap between UPDATEDLAMA and NEWLAMA.

Table 13: F1 Score of Main Results.

| Method | IL
EM | UL
EM | NL
EM | NLE
EM | FUAR
$((\mathbf{IL}), \mathbf{UL}, \mathbf{NL}) \downarrow$ |
|---|---|---|---|---|---|
| T5-Initial | **24.88** | 2.62 | 3.19 | 14.49 | - |
| T5-Vanilla | 13.11 | 11.89 | 5.84 | 22.53 | 0.68 |
| T5-RecAdam | 13.39 | 14.33 | 6.15 | 22.68 | 0.57 |
| T5-MixReview | 14.09 | 8.11 | 4.80 | 18.89 | 1.10 |
| T5-LoRA | 17.04 | **14.50** | **7.45** | **24.59** | 0.36 |
| T5-Kadapters (k=2) | 19.88 | 13.67 | 7.43 | 24.04 | 0.22 |
| T5-Kadapters (k=3) | 19.91 | 14.31 | 6.55 | 23.33 | 0.21 |
| T5-Modular | 21.35 | 12.78 | 6.94 | 24.42 | **0.17** |

frequency in each corpus of the entities that serve as the ground truths, e.g., those entities appear more in corpus $D_0$ than in $D_1$. In order to get rid of the influence of frequency of entities when analyzing the source of the EM gap, we find overlapping *Person* type answers from UL and NL, and analyze only the 67 probing sentences for both datasets each paired to one of these entities. As shown in Figure 8b, the EM on UL is still much higher than that of NL. Manually analyzing these instances, we find that the probing sentences for NL ask for relatively more *fine-grained* knowledge compared to UL, since the instances of UL by definition are overlapped cloze sentences with different answers in the corpus $D_0$ and $D_1$, that naturally make them be *coarse-grained*. For instance, the probing sentences for entity "Tim Walz" in UL and NL are "\_\_\_\_\_ is the governor of Minnesota this year." and "\_\_\_\_\_ is the governor of Minnesota calling for the evacuation of St. Paul.", respectively. We thus conjecture that the main causation of the EM gap to be UL consisting of instances requiring *coarse-grained* knowledge, which is likely to have appeared more during in $D_1$, while NL consisting of instances requiring *fine-grained* knowledge, which is expected to likely have appeared less in $D_1$.

## J  ADDITIONAL ANALYSIS OF MAIN RESULTS

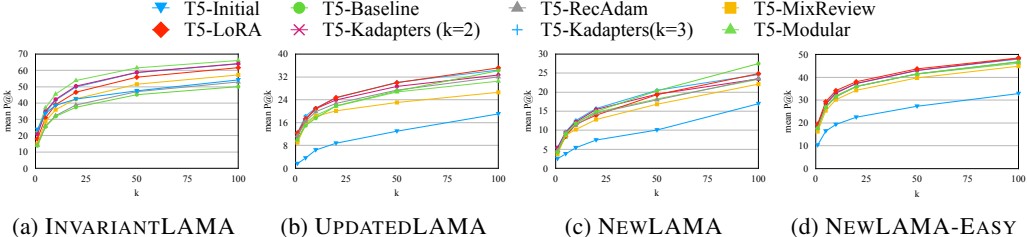

Figure 9: Mean P@k curve for CKL benchmark with varying k.

