# OpenReview forum: "Towards Continual Knowledge Learning of Language Models"
_ICLR.cc/2022/Conference — ICLR 2022 Poster_

### Official Review · Reviewer_Rjyt · 2021-11-01

**Correctness:** 3
**Technical Novelty And Significance:** 3
**Empirical Novelty And Significance:** 4
**Recommendation:** 8
**Confidence:** 4

**Main Review:**

Strengths:
In terms of novelty, the paper extended the definition of Continuous Learning to formulate Continuous Knowledge Learning that has unique challenges compared to the traditional CL. The paper also introduced a novel metric named FUAR to measure the trade-off between knowledge forgetting, update, or acquisition. This is a contribution to the field as this quantitative metric could facilitate direct comparison between models performing CKL tasks.

The paper is technically sound. Extensive experiments were conducted to benchmark the performance of various CL approaches (regularization, rehearsal, and parameter-expansion methods) on different aspects of the CKL task (retention of time-invariant knowledge, updating old knowledge, and acquiring new knowledge). In the Appendix, the authors also presented various ways to understand the model's learning process, such as the change of predicted outputs during the continued pretraining, as well as the failure analysis based on the type of probes. These methods provide more insight into the model's learning process that went beyond the plain performance scores.

Weaknesses:
Using T5, the authors showed that parameter-expansion methods have the most robust performance throughout all of the experimental settings. However, in the experiments with GPT-2 (a decoder-only model) in the Appendix, GPT2-MixReview (a rehearsal method) performs the best. Although the authors mentioned that “We leave more exploration of applying CKL methods on decoder-only models such as GPT-2 architecture as future work”, they should still mention this critical discrepancy in the main body of the paper so that the readers are aware of the context of the findings. After all, large language models take various forms and both T5 and GPT-2 are examples of large language models.

Questions:
The authors demonstrated that parameter-expansion methods have the most robust performance throughout all of the experimental settings with T-5. Since the three methods (regularization, rehearsal, and parameter expansion) are not mutually exclusive, I was wondering if the authors have tried any combination of the approaches. For instance, the rehearsal approach could be combined with the parameter-expansion method. I was wondering whether a combined approach would yield even higher performance.


**Summary Of The Paper:**

The authors formulate a new continual learning (CL) problem called Continual Knowledge Learning (CKL). Particularly, they distinguished three sub-tasks in CKL, i.e., the retention of time-invariant world knowledge, the update of old knowledge, and the acquisition of new knowledge. They also introduce a new benchmark and metric to quantify the performance of various state-of-the-art models on these sub-tasks. They find that CKL demonstrates unique challenges that are not present in previous CL setups. Critical causes of knowledge forgetting in CKL are also discussed.

**Summary Of The Review:**

The paper formulated the problem of Continual Knowledge Learning and benchmarked the performance of large language models on this task with different CL methods. The tradeoff between forgetting existing knowledge and updating old knowledge/acquiring new knowledge is quantified through a new metric, which would serve as an important optimization goal for future research. This work is a big contribution to the community and would invite more research into this topic.

---

> ### Author Response · Authors · 2021-11-15
> **Response to Reviewer Rjyt**
>
> Hello Reviewer Rjyt,
>
> Thank you for taking your time and providing us with a detailed analysis and positive feedback!! The authors want to point out that the discrepancy between different LM architectures was included when listing the main contributions of the paper in Section 1 as follows:
>
> *“... CKL methods are transferable across LM architectures despite showing a different trend in performance.”*
>
> Moreover, we are revising our submission and including the possible causes of the discrepancy of the performance trend between the encoder-decoder LM and decoder-only LM in Appendix G of the paper so that readers can be aware and explore training ever-changing LMs for decoder-only models for future work as follows:
>
> > We want to highlight that the discrepancy of the performances among the CKL methods between encoder-decoder LM (T5) and decoder-only LM (GPT-2) may not solely be on the LM architecture, but also on the learning rate and the evaluation method (light-tuning was used to evaluate GPT-2 while we evaluated T5 in a zero-shot manner). We leave further exploration of training ever-changing decoder-only LMs such as GPT-2 as future work.
>
> Regarding your question, since parameter-expansion requires freezing the parameters, we deemed it was incompatible with regularization methods (RecAdam) which regularizes the weights during continued pretraining. Also, we didn’t consider a mixture with a rehearsal-based approach since it showed worse performance than a naive approach of continually pretraining (Vanilla).
>
> **Authors’ Note**
>
> Again, the authors want to thank you for your time and positive feedback.

---

### Official Review · Reviewer_Hkbx · 2021-11-03

**Correctness:** 3
**Technical Novelty And Significance:** 2
**Empirical Novelty And Significance:** 3
**Recommendation:** 6
**Confidence:** 4

**Main Review:**

Strengths
- They provide a new benchmark and metric to measure the retention of time-invariant knowledge, updated knowledge, and new knowledge.
- Some interesting observations are provided, for example, 1) rehearsal methods do not work well in this setting (even though the reason is quite obvious because some knowledge is updated) and parameter-expansion methods achieve better results, 2) LMs are prone to more forgetting as they go through multiple CKL phases, 3)  LMs should be pretrained with just a few epochs on less duplicating data for efficiency.

Weakness
- In a real-world scenario, how can one know in advance that the new task is truly "new"? As mentioned on page 5, it is possible that the FUAR score is very large if "no gain" and "preserve knowledge". The authors did not show experiments or analysis in such a setting, where the new task has some "knowledge overlapping" with the learned tasks. To make this concern more general, I feel the measurement of "task similarity" is missing in this work.

- Can we also see perplexity as another dimension? EM scores cannot know "how bad" the prediction distribution is.

- Do you think the conclusion of these LAMA tasks is the same as other NLP downstream tasks?

- Do you think others can easily replicate your numbers? Do you run several splitting or seeds for multiple round CL settings?

- Have you tried different sizes of T5 models? Maybe the GAP between vanilla and CKL methods will be smaller given larger models.

Misc
- I have doubt on this sentence "Moreover, the effectiveness of CKL methods is much reduced in multi-phase CKL, shown by the decrease of the gap between the FUAR of T5-Vanilla and the best performing CKL method in the scenario of two-phase and one phase, which is 0.92 and 0.46, respectively." Isn't it only prove that T5-Vanilla can learn better FUAR scores SMALL-P1-->  SMALL-P2 than SMALL-P1+  SMALL-P2 because you have similar FUAR scores for T5-Kadapters?

**Summary Of The Paper:**

- The paper is about continuous learning for language models. The authors leverage existing LAMA tasks and collect a new test benchmark with updated information and new information.
- They investigate several existing CL algorithms, and they propose a new metric called FUAR to measure trade-off between forgotten time-invariant knowledge and updated or newly acquired knowledge.
- They provided some findings on their continuous LM learning.

**Summary Of The Review:**

This work is quite insightful for us to understand more about how LM continuous learning works, although I think more experiments could be beneficial as I mentioned in the weakness section. If we can make sure the numbers from this paper are reproducible and comparable, then I think it could be a good testbed for future research in this direction.

---

> ### Author Response · Authors · 2021-11-17
> **Response to Reviewer Hkbx**
>
> Hello Reviewer Hkbx,
>
> Thank you for your valuable time on the extensive analysis highlighting the strengths of the paper! The authors want to kindly highlight some key points in the paper that may address the weaknesses that you have pointed out.
>
> > Weakness 1: In a real-world scenario, how can one know in advance that the new task is truly "new"? As mentioned on page 5, it is possible that the FUAR score is very large if "no gain" and "preserve knowledge". The authors did not show experiments or analysis in such a setting, where the new task has some "knowledge overlapping" with the learned tasks. To make this concern more general, I feel the measurement of "task similarity" is missing in this work.
>
> Thank you for your comment. The authors agree that in real-world scenarios, tasks may not be mutually exclusive in terms of requiring time-invariant, new, and updated world knowledge. Most real-world tasks may consist of a mixture of instances that require different knowledge criteria. Our work can be seen as an evaluation benchmark dissecting such a task to evaluate methodologies for training ever-changing LMs. For example, evaluating a probing task (InvariantLAMA + UpdatedLAMA + NewLAMA) without dividing each instance mutually exclusively will result in a performance where it is impossible to evaluate whether the performance is due to effectively retaining time-invariant knowledge or successfully updating or acquiring new knowledge.
>
> > Weakness 2: Can we also see perplexity as another dimension? EM scores cannot know "how bad" the prediction distribution is.
>
> Thank you for your comment. The authors agree that perplexity can be seen as another dimension. However, it is not straightforward to compute perplexity on a corpus with an encoder-decoder LM since the input-output sequences are different ( salient-span masking proposed by (Guu, et.al. 2020) was used for pretraining objective). Instead, responding to your feedback, we are including the F1 score computed for the main experiments in the Appendix J as follows :
>
> |       Method       |   IL  |   UL  |  NL  |  NLE  |      |
> |:------------------:|:-----:|:-----:|:----:|:-----:|------|
> |                    |   F1  |   F1  |  F1  |   F1  | FUAR |
> |     T5-Initial     | **24.88** |  2.62 | 3.19 | 14.49 |      |
> |     T5-Baseline    | 13.11 | 11.89 | 5.84 | 22.53 | 0.68 |
> |     T5-RecAdam     | 13.39 | 14.33 | 6.15 | 22.68 | 0.57 |
> |    T5-MixReview    | 14.09 |  8.11 |  4.8 | 18.89 | 1.10 |
> |       T5-LoRA      | 17.04 |  **14.5** | **7.45** | **24.59** | 0.36 |
> | T5-Kadapters (k=2) | 19.88 | 13.67 | *7.43* | 24.04 | 0.22 |
> | T5-Kadapters (k=3) | 19.91 | *14.31* | 6.55 | 23.33 | *0.21* |
> | T5-Modular         | *21.35* | 12.78 | 6.94 | *24.42* | **0.17** |
>
> best is **bolded**, second best is *italicized*.
>
> Moreover, in order to further analyze the prediction distribution, we are adding the top-k mean precision accuracy (k=1,5,10,20,50,100) for the main experiments in Appendix J as well.
>
> > Weakness 3: Do you think the conclusion of these LAMA tasks is the same as other NLP downstream tasks?
>
> Thank you for your comment. The authors agree that other NLP downstream tasks are the fundamental end goal of training ever-changing LMs (slot-filling is not the main use-case of LMs in the real-world). We provided experiments performed on all of KILT tasks (Petroni, et.al. 2020) after CKL with all the CKL methods in Appendix F and showed that performing downstream tasks show the same trend among methods for tasks requiring world knowledge from D_0 (initial corpus). For downstream tasks requiring new/updated knowledge, we only showed the probing performances in this paper and provide the corresponding QA pairs for each CKL task so that future work can address effectively utilizing the knowledge stored in the implicit parameters of language models for knowledge-intensive tasks without the test-train overlap problem (Lewis, et.al. 2020), which we deemed was out-of-scope with respect to this research.
>
> > Weakness 4: Do you think others can easily replicate your numbers? Do you run several splitting or seeds for multiple round CL settings?
>
> Thank you for your comments. Pretraining large LMs requires heavy computational resources and thus we were unable to provide multiple pretraining runs with random seeds. However, we provided experimental results on different corpora and in two phases (Section 5.2) and showed the same trend among the baseline methods, which was one of our main research objectives. We also plan to release the model checkpoints for the main experiment so that others can easily replicate the evaluation result numbers on the CKL benchmark.

---

> > ### Author Response · Authors · 2021-11-17
> > **Response to Reviewer Hkbx (continued)**
> >
> > > Weakness 5: Have you tried different sizes of T5 models? Maybe the GAP between vanilla and CKL methods will be smaller given larger models.
> >
> > Thank you for your comment. The authors agree that experimental results on bigger variants of T5 (T5-3B, T5-11B) will be much helpful for analyzing the full extent of the effectiveness of CKL methods. We believe how the scaling law of LM affects CKL is an interesting topic that should be explored in future work.
> >
> > > Misc 1: I have doubt on this sentence "Moreover, the effectiveness of CKL methods is much reduced in multi-phase CKL, shown by the decrease of the gap between the FUAR of T5-Vanilla and the best performing CKL method in the scenario of two-phase and one phase, which is 0.92 and 0.46, respectively." Isn't it only prove that T5-Vanilla can learn better FUAR scores SMALL-P1--> SMALL-P2 than SMALL-P1+ SMALL-P2 because you have similar FUAR scores for T5-Kadapters?
> >
> > Thank you for your correction. We agree that the gap in FUAR score was due to the decrease of T5-Vanilla instead of the increase of T5-Kadapters. We are deleting the following sentence and the similar remarks in the revision.
> >
> > **Authors’ Note**
> >
> > To this end, we hope the responses have answered the reviewer’s concerns. We are open to more discussion!
> >
> > **Reference**
> >
> > Guu, K., Lee, K., Tung, Z., Pasupat, P., & Chang, M. W. (2020). Realm: Retrieval-augmented language model pre-training. arXiv preprint arXiv:2002.08909.
> >
> > Lewis, P., Stenetorp, P., & Riedel, S. (2020). Question and answer test-train overlap in open-domain question answering datasets. arXiv preprint arXiv:2008.02637.
> >
> > Petroni, F., Piktus, A., Fan, A., Lewis, P., Yazdani, M., De Cao, N., ... & Riedel, S. (2020). Kilt: a benchmark for knowledge intensive language tasks. arXiv preprint arXiv:2009.02252.

---

### Official Review · Reviewer_ELX1 · 2021-11-03

**Correctness:** 3
**Technical Novelty And Significance:** 3
**Empirical Novelty And Significance:** 3
**Recommendation:** 6
**Confidence:** 4

**Main Review:**

Strengths:
1. The proposed continual knowledge learning problem is quite interesting and important.
2. The benchmark is useful and the proposed FUAR metric is technically sound.

Weaknesses:
1. The paper only performs experiments with an encoder-decoder model (T5). The experimental results will be more convincing if more pre-trained language models (such as GPT) are included. For example, we can explore the ability of different PLMs to avoid catastrophic forgetting and to acquire new knowledge while preserving invariant knowledge.
2. Consisting with the traditional setting of CL, the paper also creates a setting for multiple CKL phases. However, only a two-phase setting is considered. More experiments can be explored, such as five-phase or controlling the differences in the distribution of data at different phases.
3. The experimental findings in this paper are somewhat trivial.


**Summary Of The Paper:**

The paper studies continual knowledge learning of language models, which is an interesting and important problem. Particularly, a new benchmark and a metric are introduced to quantify the retention of time-invariant world knowledge, the update of outdated knowledge, and the acquisition of new knowledge. To establish baselines for the CKL benchmark and validate the rationality of the proposed benchmark and metric, the author conducts extensive experiments with a pre-trained encoder-decoder model (T5) based on various training methodologies including regularization, rehearsal, and parameter expansion methods.

The paper is well organized and easy to follow. The proposed continual knowledge learning problem is quite interesting and important. The FUAR metric is also technically sound. The authors also conduct comprehensive experiments to verify the rationality of the proposed benchmark under the various settings.

**Summary Of The Review:**

The paper studies an interesting problem. The proposed benchmark and metric are technically sound. However, there are some concerns about experimental settings.

---

> ### Author Response · Authors · 2021-11-15
> **Response to Reviewer ELX1**
>
> Hello Reviewer ELX1,
>
> Thank you for your valuable time in reviewing our paper and providing an extensive analysis highlighting the strengths of the paper! The authors want to kindly highlight some key points in the paper that may address the weaknesses that you have pointed out.
>
> > Weakness 1: The paper only performs experiments with an encoder-decoder model (T5). The experimental results will be more convincing if more pre-trained language models (such as GPT) are included. For example, we can explore the ability of different PLMs to avoid catastrophic forgetting and to acquire new knowledge while preserving invariant knowledge.
>
> Thank you for your comment. The authors agree that different LM architecture has to be considered. We therefore explored decoder-only LM architecture (GPT-2) in “Appendix G Exploring How CKL Methods Transfer Across LM Architectures” where we applied all of the corresponding baseline methods during continued pretraining and evaluated their effectiveness with the CKL benchmark.
>
> > Weakness 2: Consisting with the traditional setting of CL, the paper also creates a setting for multiple CKL phases. However, only a two-phase setting is considered. More experiments can be explored, such as five-phase or controlling the differences in the distribution of data at different phases.
>
> Thank you for your comment. In realistic scenarios, LMs may need to be updated regularly (once a week/month) and the authors agree that the limitation of this paper is only considering one & two phases. Moreover, as the interval between updates may vary which will result in the difference of data distribution (definitely in terms of size), we strongly believe that the next future work towards training ever-changing LMs should address multiple updates with streaming corpora of different sizes and distributions. This paper can be considered as the starting point for such research.
>
> > Weakness 3: The experimental findings in this paper are somewhat trivial.
>
> Thank you for your comment. The authors would like to emphasize that we are the first to define the continual knowledge learning problem, explore methodologies that are natural baselines, propose an intuitive metric measuring the overall trade-off between forgetting and new knowledge gained, and provide the first benchmark for evaluating methods for training ever-changing LMs.
>
> **Authors’ Note**
>
> To this end, the authors hope the responses have addressed most of the reviewer’s concerns and are open to more discussion!

---

> > ### Comment · Reviewer_ELX1 · 2021-12-02
> > **Response to Rebuttal**
> >
> > Thank you for answering my questions. I stick to my rating since the rebuttal doesn't fully address my concerns.

---

### Official Review · Reviewer_U9Hk · 2021-11-04

**Correctness:** 3
**Technical Novelty And Significance:** 2
**Empirical Novelty And Significance:** 2
**Recommendation:** 3
**Confidence:** 5

**Main Review:**

Strengths:
- The problem of CLK itself is an important task and more realistic to downstream knowledge-intensive applications.
- I like the clear separation of the types of knowledge: time-invariant (to keep), outdated (to remove), new (to inject), as well as their collected datasets for reflecting the three types of knowledge update.
- Different types of baseline methods are covered and compared with analysis.

Weakness:
- The formulation of the CKL problem is overly simplified. It basically only considers a single corpus (D_1) for updating the knowledge of previously learned LMs. A general setup should consider a streaming version of D_1 and make it a sequence of sub-corpus (D_1, D_2, ..., D_T) that arrive at different time steps. Also, the associated tasks in UpdatedLAMA and NewLAMA should reflect such a time series --- that is, the streaming version of the current probing tasks. The current formulation described in Section 3.1 only has a single time step. It's more like an offline learning problem with the forgetting constraints but not an (online) continual learning problem. The proposed setup is thus a bit far from the motivation of studying CKL --- maintaining an ever-changing LM.
- The experiments are very limited to the LAMA probing which does not necessarily connect to real downstream applications of these LMs. It's also hard to justify whether such methods can maintain performance in general NLP tasks. The argument about the KILT experiments seems to only focus on testing the retention but not about the new/updated knowledge about D_1.
- The analysis of the CKL methods is not deep enough. How do these methods work and why do some outperform others? How do we know if an arbitrary new fact conflict with the existing knowledge or not on the fly? How do you define such conflicts properly? Say you have a sentence in D_0 "Cristiano Ronaldo plays for XXX in 2010", and you have another sentence in D_1 "Cristiano Ronaldo plays for YYY now." In the current problem setup, how will such "conflict" be defined?
- The new findings are not particularly non-trivial.
- The presentation and the writing of the paper can be further improved with more illustrative examples and case studies for readers to qualitatively see the problem setup and the differences between these methods.

**Summary Of The Paper:**

This paper presents a new continual learning problem setup: continual knowledge learning (CKL) and constructs an associated benchmark resource. The benchmark is based on slot filling-based knowledge probing tasks (i.e., the LAMA analysis). The authors show the empirical performance of some existing CL methods, ranging from regularization, rehearsal, and parameter expansion. And they show a few findings based on their experimental results, e.g., learning rate can be sensitive to balance the tradeoff between forgetting and learning new knowledge, and CKL methods might have transferrable performance across different LMs. (e.g., T5 and GPT).

**Summary Of The Review:**

This paper is a pilot study of an improtant problem (continual knowledge learning of LMs), but the problem formulation is overly simple and there are still many important yet missing points in both data construction and experiments. Also, there are no much insightful and non-trivial findings with deep analysis of existing methods. Please find more details above.

---

> ### Author Response · Authors · 2021-11-15
> **Response to Reviewer U9Hk**
>
> Hello Reviewer U9Hk,
>
> Thank you for your valuable time in reviewing our paper. The authors respectively argue that most of the mentioned weaknesses were addressed in the original submission.
>
> > Weakness 1: The formulation of the CKL problem is overly simplified. It basically only considers a single corpus (D_1) for updating the knowledge of previously learned LMs. A general setup should consider a streaming version of D_1 and make it a sequence of sub-corpus (D_1, D_2, ..., D_T) that arrive at different time steps. Also, the associated tasks in UpdatedLAMA and NewLAMA should reflect such a time series --- that is, the streaming version of the current probing tasks. The current formulation described in Section 3.1 only has a single time step. It's more like an offline learning problem with the forgetting constraints but not an (online) continual learning problem. The proposed setup is thus a bit far from the motivation of studying CKL --- maintaining an ever-changing LM.
>
> The authors agree that in order for training truly ever-changing LMs, we need multiple streams of corpora (D_1, D_2, … D_T). In Section 5.2  we provide a streaming version of D_1 (05.2020-11.2020 & 11.2020 - 04.2021) and also provide the corresponding evaluation dataset NewLAMA-Easy (NLE) divided into two probing tasks, NLE-P1, NLE-P2,  according to the streaming time-series. We perform extensive experiments in such setting and show that it is harder to keep time-invariant knowledge during multiple phases and also highlight the weakness of parameter-expansion methods since new parameters have to be added during each LM update.
>
> > Weakness 2: The experiments are very limited to the LAMA probing which does not necessarily connect to real downstream applications of these LMs. It's also hard to justify whether such methods can maintain performance in general NLP tasks. The argument about the KILT experiments seems to only focus on testing the retention but not about the new/updated knowledge about D_1.
>
> Thank you for your comment. The authors agree that downstream tasks for testing the new/updated knowledge have not been shown in this paper. This is because it has recently been highlighted that a large portion of the performance gain from performing downstream tasks utilizing implicit knowledge (e.g., Closed-book QA) after pretraining comes from the test-train overlap (Lewis, et.al. 2020) instead of actually utilizing the knowledge stored in the implicit parameters of the LM, and thus we have deemed that the effective utilization of the newly updated/gained knowledge from pretraining for downstream task would be a totally new research problem and out of scope with respect to our research. However, during the data construction phase, we constructed the QA version of all of the probing tasks for new/updated knowledge and provided it together with the CKL benchmark so that future research can explore methods utilizing the new/updated knowledge gained for downstream tasks.
>
> > Weakness 3: The analysis of the CKL methods is not deep enough. How do these methods work and why do some outperform others? How do we know if an arbitrary new fact conflict with the existing knowledge or not on the fly? How do you define such conflicts properly? Say you have a sentence in D_0 "Cristiano Ronaldo plays for XXX in 2010", and you have another sentence in D_1 "Cristiano Ronaldo plays for YYY now." In the current problem setup, how will such "conflict" be defined?
>
> Thank you for your comment. The authors provided an explanation of how each CKL method works in Appendix A.2 where we describe how each method overcomes the difference between CKL and previous continual learning setups. We also provided how we define “fact conflict” in Appendix B where we go in-depth on the data construction process and how we define “updated” and “new” knowledge. For probing “updated” facts (or fact conflicts), annotators made sure that the answer for the probe found in D_0 was different from the answer found in D_1. For example, we fixed a probe “Cristiano Ronaldo plays for _____.” as a probe instance in UpdatedLAMA and made sure that different answers can be found in D_0 and D_1. Moreover, we are adding the following sentence in the revised draft according to your feedback to include how we handled cases where the time-stamps are fixed in Section 3.1:
>
> *'Also, we classify instances where the time-stamps are fixed such as "Cristiano Ronaldo played for _____ in 2010." as time-invariant.'*
>
> > Weakness 4: The new findings are not particularly non-trivial.
>
> Thank you for your comment. The authors want to emphasize that we are the first to define the continual knowledge learning problem (which we argued to be an important problem), explore various methodologies, propose an intuitive metric measuring the overall trade-off between forgetting and new knowledge gained, and provide the first benchmark for evaluating methods for training ever-changing LMs.

---

> > ### Author Response · Authors · 2021-11-15
> > **Response to Reviewer U9Hk (continued)**
> >
> >
> > > Weakness 5: The presentation and the writing of the paper can be further improved with more illustrative examples and case studies for readers to qualitatively see the problem setup and the differences between these methods.
> >
> > Thank you for your comment. The authors provided illustrative examples of the CKL benchmark in Table 6 in Appendix B.3 Dataset Statistics and Examples. We also showed the prediction output (as training progresses) of baseline methods (vanilla & modular) in each CKL task so that the readers can qualitatively access the problem setup and how applying CKL methods helps LMs retain time-invariant knowledge while effectively updating and gaining new knowledge as the LMs are continually pretrained (Appendix H). Lastly, we also showed a distribution of probe types depending on the entity type and how each type contributes to the performance of each task in Appendix I.
> >
> > **Authors’ note**
> >
> > The authors hope our responses have addressed the reviewer’s concerns and hope the reviewer could reconsider his/her comments and score recommendation.
> >
> > **Reference**
> >
> > Lewis, P., Stenetorp, P., & Riedel, S. (2020). Question and answer test-train overlap in open-domain question answering datasets. arXiv preprint arXiv:2008.02637.

---

### Decision · Program_Chairs · 2022-01-20

**Decision:**

Accept (Poster)

**Comment:**

The paper introduces the problem of continual knowledge (language) learning. The authors point out the interesting duality between continual learning and knowledge learning where: in knowledge learning one must avoid forgetting time-invariant knowledge (avoid forgetting in CL), be able to acquire new knowledge (learn new tasks in CL), and replace outdated knowledge (a form of forgetting and re-learning or adaptation). In their paper, the authors develop an initial benchmark for the task along with a set of baselines and provide empirical studies.

The initial reviews were quite mixed. The reviewers seem to agree this work studies an interesting and fairly novel direction for continual learning of language. However, the reviewers did not agree on whether this initial stab at the problem was "enough." In particular, reviewer U9Hk argues that the formulation is "oversimplified" and the current experiments are limiting.

After the discussion, the reviewers remained split with one high score (8), two borderline accepts (3), and one reject. So three reviewers believe that this manuscript is already a good contribution. The fourth reviewer disagrees, but the authors provided clear and convincing responses to many of their comments (and point to results already available in the appendix).

Overall, this is a clear and reasonable first step considering this paper proposes a new CL problem. The reviewers and I believe that this is interesting and rigorous enough to be impactful and to warrant follow-up works. As a result, I'm happy to recommend acceptance. I imagine that if the community demonstrates interest in this line of work, there will be work both on methodologies to improve the proposed baselines, but also work proposing extensions to the problem in line with some of the comments of reviewer U9Hk.

In preparing their camera-ready version I strongly encourage the authors to take into account the suggestions of the reviewers and your replies. In particular, your discussion regarding encoder-decoder and decoder-only LMs and the associated results would be good to discuss in the main text (even if the full results are in the appendix).